# DATASET INFERENCE:
# OWNERSHIP RESOLUTION IN MACHINE LEARNING

**Pratyush Maini**
IIT Delhi*
pratyush.maini@gmail.com

**Mohammad Yaghini, Nicolas Papernot**
University of Toronto and Vector Institute
{mohammad.yaghini,nicolas.papernot}@utoronto.ca

## ABSTRACT

With increasingly more data and computation involved in their training, machine learning models constitute valuable intellectual property. This has spurred interest in model stealing, which is made more practical by advances in learning with partial, little, or no supervision. Existing defenses focus on inserting unique watermarks in a model's decision surface, but this is insufficient: the watermarks are not sampled from the training distribution and thus are not always preserved during model stealing. In this paper, we make the key observation that knowledge contained in the stolen model's training set is what is common to all stolen copies. The adversary's goal, irrespective of the attack employed, is always to extract this knowledge or its by-products. This gives the original model's owner a strong advantage over the adversary: model owners have access to the original training data. We thus introduce *dataset inference*, the process of identifying whether a suspected model copy has private knowledge from the original model's dataset, as a defense against model stealing. We develop an approach for dataset inference that combines statistical testing with the ability to estimate the distance of multiple data points to the decision boundary. Our experiments on CIFAR10, SVHN, CIFAR100 and ImageNet show that model owners can claim with confidence greater than 99% that their model (or dataset as a matter of fact) was stolen, despite only exposing 50 of the stolen model's training points. Dataset inference defends against state-of-the-art attacks even when the adversary is adaptive. Unlike prior work, it does not require retraining or overfitting the defended model.[1]

## 1 INTRODUCTION

Machine learning models have increasingly many parameters (Brown et al., 2020; Kolesnikov et al., 2019), requiring larger datasets and significant investment of resources. For example, OpenAI's development of GPT-3 is estimated to have cost over USD 4 million (Li, 2020). Yet, models are often exposed to the public to provide services such as machine translation (Wu et al., 2016) or image recognition (Wu et al., 2019). This gives adversaries an incentive to steal models via the exposed interfaces using model extraction. This threat raises a question of ownership resolution: *how can an owner prove that another suspect model stole their intellectual property?* Specifically, we aim to determine whether a potentially stolen model was derived from an owner's model or dataset.

An adversary may derive and steal intellectual property from a victim in many ways. A prominent way is **(1)** model extraction (Tramèr et al., 2016), where the adversary exploits access to a model's **(1.a)** prediction vectors (e.g., through an API) to reproduce a copy of the model at a lower cost than what is incurred in developing it. Perhaps less directly, **(1.b)** the adversary could also use the victim model as a labeling oracle to train their model on an initially unlabeled dataset obtained either from a public source or collected by the adversary. In a more extreme threat model, **(2)** the adversary could also get access to the dataset itself which was used to train the victim model and train their own model by either **(2.a)** distilling the victim model, or **(2.b)** training from scratch altogether. Finally, adversaries may gain **(3)** complete access to the victim model, but not the dataset. This may happen when a victim wishes to open-source their work for academic purposes but disallows its

---

*Work done while an intern at the University of Toronto and Vector Institute

[1]Code and models for reproducing our work can be found at github.com/cleverhans-lab/dataset-inference

commercialization, or simply via insider-access. The adversary may **(3.a)** fine-tune over the victim model, or **(3.b)** use the victim for data-free distillation (Fang et al., 2019).

Preventing all forms of model stealing is impossible without decreasing model accuracy for legitimate users: model extraction adversaries can obfuscate malicious queries as legitimate ones from the expected distribution. Most prior efforts thus focus on watermarking models before deployment. Rather than preventing model stealing, they aim to detect theft by allowing the victim to claim ownership by verifying that a suspect model responds with the expected outputs on watermarked inputs. This strategy not only requires re-training and decreases model accuracy, it can also be vulnerable to adaptive attacks that lessen the impact of watermarks on the decision surface during extraction. Thus, recent work that has managed to prevail (Yang et al., 2019) despite distillation (Hinton et al., 2015) or extraction (Jia et al., 2020), has suffered a trade-off in model performance.

In our work, we make the key observation that all stolen models necessarily contain direct or indirect information from the victim model's training set. This holds regardless of how the adversary gained access to the stolen model. This leads us to propose a fundamentally different defense strategy: we identify stolen models because they possess knowledge contained in the private training set of the victim. Indeed, a successful model extraction attack will distill the victim's knowledge of its training data into the stolen copy. Hence, we propose to identify stolen copies by showing that they were trained (at least partially and indirectly) on the same dataset as the victim.

We call this process *dataset inference* (DI). In particular, we find that stolen models are more confident about points in the victim model's training set than on a random point drawn from the task distribution. The more an adversary interacts with the victim model to steal it, the easier it will be to claim ownership by distinguishing the stolen model's behavior on the victim model's training set. We distinguish a model's behavior on its training data from other subsets of data by measuring the 'prediction certainty' of any data point: the margin of a given data point to neighbouring classes.

At its core, DI builds on the premise of input memorization, albeit weak. One might think that DI succeeds only for models trained on small datasets when overfitting is likely. Surprisingly, in practice, we find that even models trained on ImageNet end up memorizing training data in some form.

Among related work discussed in § 2, distinguishing a classifier's behavior on examples from its train and test sets is closest to membership inference (Shokri et al., 2017). Membership inference (MI) is an attack predicting whether *individual* examples were used to train a model or not. *Dataset inference* flips this situation and exploits information asymmetry: the potential victim of model theft is now the one testing for membership and naturally has access to the training data. Whereas MI typically requires a large train-test gap because such a setting allows a greater distinction between *individual* points in and out the training set (Yeom et al., 2018; Choo et al., 2020), dataset inference succeeds even when the defender has slightly better than random chance of guessing membership correctly; because the victim aggregates the result of DI over multiple points from the training set.

In summary, our contributions are:

- We introduce *dataset inference* as a general framework for ownership resolution in machine learning. Our key observation is that knowledge of the training set leads to information asymmetry which advantages legitimate model owners when resolving ownership.
- We theoretically show on a linear model that the success of MI decreases with the size of the training set (as overfitting decreases), whereas DI is independent of the same. Despite the failure of MI on a binary classification task, DI still succeeds with high probability.
- We propose two different methods to characterize training vs. test behavior: targeted adversarial attacks in the white-box setting, and a novel 'Blind Walk' method for the black-box label-only setting. We then create a concise embedding of each data point that is fed to a confidence regressor to distinguish between points inside and outside a model's training set. Hypothesis testing then returns the final ownership claim.
- Unlike prior efforts, our method not only helps defend ML services against model extraction attacks, but also in extreme scenarios such as complete theft of the victim's model or training data. In § 7, we also introduce and evaluate our approach against adaptive attacks.
- We evaluate our method on the CIFAR10, SVHN, CIFAR100 and ImageNet datasets and obtain greater than 99% confidence in detecting model or data theft via the threat models studied in this work, by exposing as low as 50 *random* samples from our private dataset.

We remark that dataset inference applies beyond intellectual property issues. For example, Song & Shmatikov (2019) showed that models trained for gender classification also learn features predictive of ethnicity. This raises ethical concerns, and dataset inference could assess whether a sensitive dataset was used by a model developer for different purposes than stated at data collection time.

## 2  RELATED WORK

**Model Extraction.**  Model extraction (Tramèr et al., 2016; Jagielski et al., 2020; Truong et al., 2021) is the process where an adversary tries to steal a copy of a machine learning model, that may have been remotely deployed (such as over a prediction API). Depending on the level of access provided by the prediction APIs, model extraction may be performed by only using the labels (Chandrasekaran et al., 2019; Correia-Silva et al., 2018) or the entire prediction logits of the deployed service (Orekondy et al., 2018). Model extraction has seen a cycle of attacks and defenses. Once an adversary has knowledge of the defense strategy adopted by the victim, they adaptively modify the attack to circumvent that defense (see watermarking). Model extraction can also be a reconnaissance step used to prepare for further attacks, e.g., finding adversarial examples (Papernot et al., 2017; Shumailov et al., 2020).

**Watermarking.**  Since Uchida et al. (2017) embedded watermarks into neural networks and Adi et al. (2018) used them as signatures to claim possession, watermarks have been widely adopted as a way to resolve ownership claims. The underlying idea is to manipulate the model to learn information other than that from the true data distribution, and use this knowledge for verification afterwards. This strategy not only requires new training procedures and decreases the model's accuracy (Jia et al., 2020), but is also vulnerable to adaptive attacks that lessen the impact of watermarks on the model's decision surface during extraction (Liu et al., 2018; Chen et al., 2019; Wang et al., 2019; Shafieinejad et al., 2019).

**Membership Inference.**  Shokri et al. (2017) train a number of *shadow* classifiers on confidence scores produced by the target model with labels indicating whether samples came from the training or testing set. MI attacks are shown to work in white- (Leino & Fredrikson, 2020; Sablayrolles et al., 2019) as well as black-box scenarios against various target models including generative models (Hayes et al., 2019). Yeom et al. (2018) explore overfitting as the root cause of MI vulnerability. Choo et al. (2020) show that MI can succeed even in scenarios when the victim only provides labels.

**Out of Distribution Detection.**  Liang et al. (2017) and Lee et al. (2018) measure model performance on modifying an input to find if a sample is in or out-of-distribution. The premise is that in-distribution samples are easier to manipulate, whereas out-of-distribution samples require more work. In contrast, our work solves a much more challenging problem: the dataset distribution may be the same, but can we still identify which of the datasets was used for training?

## 3  THREAT MODEL AND DEFINITION OF DATASET INFERENCE

Consider a victim $\mathcal{V}$ who trains a model $f_{\mathcal{V}}$ on their private data $S_{\mathcal{V}} \subseteq \mathcal{K}_{\mathcal{V}}$, where $\mathcal{K}_{\mathcal{V}}$ represents the private knowledge of $\mathcal{V}$. While $\mathcal{K}_{\mathcal{V}}$ is an abstract concept that can not be concretely defined, the private dataset $S_{\mathcal{V}}$ represents a definite part of the victim's knowledge that can be formalized. An adversary $\mathcal{A}_*$ may gain access to a subset of $\mathcal{K}_{\mathcal{V}}$ and use it to train its own model $f_{\mathcal{A}_*}$. $\mathcal{V}$ suspects theft, and would like to prove that $f_{\mathcal{A}_*}$ is indeed a copy of $f_{\mathcal{V}}$. Hence, $\mathcal{V}$ employs *dataset inference* on $f_{\mathcal{A}_*}$ to determine if a subset of their private knowledge $\mathcal{K} \subseteq \mathcal{K}_{\mathcal{V}}$ was used to train $f_{\mathcal{A}_*}$. We formally define the victim and their dataset inference experiment below.

**Definition 1 (Dataset Inferring Victim $\mathcal{V}(f, \alpha, m)$)** *Let $\mathcal{V} : \mathcal{F} \times [0, 1] \times \mathbb{N} \mapsto \{1, \emptyset\}$ be a victim with private access to $S_{\mathcal{V}} \subseteq \mathcal{K}_{\mathcal{V}}$, where $\mathcal{F}$ represents the set of all classifiers trained on samples from a data distribution $\mathcal{D}$. Given classifier $f$, $\mathcal{V}$ can reveal at most $m$ samples from $S_{\mathcal{V}}$ to either conclusively prove that a subset of their private knowledge $\mathcal{K} \subset \mathcal{K}_{\mathcal{V}}$ has been used in the training of $f$ with a Type-I error (FPR) $< \alpha$, or return an inconclusive result $\emptyset$.*

**Definition 2 (Dataset Inference Experiment $Exp^{DI}(\mathcal{V}, m, \alpha, S_{\mathcal{V}}, \mathcal{D})$)** *Let $\mathcal{F}$ be as in Definition 1, and assume $\mathcal{F}_{\mathcal{V}}$ to be the set of all classifiers trained on the victim's private dataset $S_{\mathcal{V}} \sim \mathcal{D}$, and $m$ a natural number. The dataset inference experiment follows:*

1. *Choose* $b \leftarrow \{0, 1\}$ *uniformly at random.*
2. $f_{\mathcal{A}_*} = f \sim \mathcal{F}_{\mathcal{V}}$ *if* $b = 1$*; else* $f_{\mathcal{A}_*} = f \sim \mathcal{F}$
3. $Exp^{DI}(\mathcal{V}, m, \alpha, S_{\mathcal{V}}, \mathcal{D}) = \begin{cases} 1 & \text{if } \mathcal{V}(f_{\mathcal{A}_*}, \alpha, m) = 1 \text{ and } b = 1 \\ 0 & \text{otherwise} \end{cases}$

## 4 THEORETICAL MOTIVATION

*Dataset Inference* (DI) aims to leverage the disparity in the response of an ML model to inputs that it saw during training time, versus those that it did not. We call this response 'prediction margin'. In § 4, we introduce our theoretical framework. In § 4.1, we quantify the difference in the expected response of a model to any point in the training and test set. Finally, in § 4.2 we describe how DI succeeds with high probability in this setting, while membership inference (MI) fails.

**Setup.** Consider a data distribution $\mathcal{D}$, such that any input-label pair $(\mathbf{x}, y)$ can be described as:

$$y \sim \{-1, +1\}; \quad \mathbf{x_1} = y \cdot \mathbf{u} \in \mathbb{R}^K, \quad \mathbf{x_2} \sim \mathcal{N}(0, \sigma^2 I) \in \mathbb{R}^D \tag{1}$$

where $\mathbf{x} = (\mathbf{x_1}, \mathbf{x_2}) \in \mathbb{R}^{K+D}$ and $\mathbf{u} \in \mathbb{R}^K$ is a fixed vector. Observe that the last $D$ dimensions of $\mathbf{x}$ represent Gaussian noise (with var. $\sigma^2$) having no correlation to the correct label. However, the first $K$ dimensions are sufficient to separate inputs from classes $\{-1, +1\}$ (Nagarajan & Kolter, 2019). $S \sim \mathcal{D}$, s.t. $|S| = m$ represents the private training set of a model with $m$ distinct training examples.

**Architecture.** We consider the scenario of classifying the input distribution using a linear classifier, $f$, with weights $\mathbf{w} = (\mathbf{w_1}, \mathbf{w_2})$, such that for any input $\mathbf{x}$: $f(\mathbf{x}) = \mathbf{w_1} \cdot \mathbf{x_1} + \mathbf{w_2} \cdot \mathbf{x_2}$. And the final classification decision is $\text{sgn}(f(\mathbf{x}))$. While we only discuss the case of a linear network in this analysis, the success of DI only increases with the number of parameters in a machine learning model, as is the case for MI (Yeom et al., 2018). This, in effect makes the following analysis a stronger result to prove. Prior works have also argued how over-parametrized deep learning networks memorize training points (Zhang et al., 2016; Feldman, 2019). At its core, DI builds on the premise of (weak) input memorization. Results on DNNs are discussed in § 7.

### 4.1 PREDICTION MARGIN

In our work, we use 'prediction margin' to imply the confidence of a machine learning model of its prediction. In other words, we try to capture the robustness of a model's prediction under uncertainty, which is equivalent to viewing the local landscape of a machine learning model. For the purpose of the theoretical analysis, it is convenient to define it as the margin of a data point from the decision boundary ($y \cdot f(\mathbf{x})$). As we scale our method to deep networks in the empirical evaluation, we will describe alternative methods of measuring the 'prediction margin' in multi-class settings.

**Theorem 1 (Train-Test Margin)** *Given a linear classifier $f$ trained to classify inputs $(\mathbf{x}, y) \in S$ (training set), the difference in the expected prediction margin for samples in $S$ and $\mathcal{D}$ is given by $\mathbb{E}_{(\mathbf{x},y) \sim S} [y \cdot f(\mathbf{x})] - \mathbb{E}_{(\mathbf{x},y) \sim \mathcal{D}} [y \cdot f(\mathbf{x})] = D\sigma^2$, where $\sigma^2$ is the Gaussian noise variance as in (1).*

The proof (Appendix A.2) first calculates the weights of the learned classifier $f$ by assuming that it is trained using gradient descent with a fixed learning rate, and viewing all training points exactly once. We then analyze the expected margin for data points included in training or not.

### 4.2 DATASET INFERENCE V/S MEMBERSHIP INFERENCE

We now show how MI fails to distinguish between train and test samples in the same setting. This happens because an adversary has to make a decision about the presence of a *given* data point in the training set by querying a *single* point. However, DI succeeds with high probability in the same setting because it aggregates signal over multiple data points. We note that the statistical differences between the 'prediction margin' of training and test data points in § 4.1 are only known when we calculate an expectation over multiple samples.

**Failure of Membership Inference.** Consider a membership inferring adversary $\mathcal{M}$ that has no knowledge of the victim's training data $S$, but has domain knowledge such as the publicly available data distribution $\mathcal{D}$. Define $\mathcal{M}(\mathbf{x}, f)$ as the adversary's decision function to predict whether $\mathbf{x}$ belongs to $S$. Let $\mathcal{R}$ represent a distribution that uniformly at random samples from either $S(b = 1)$ or $\mathcal{D}(b = 0)$. Then, $\mathcal{M}$ makes a membership decision about $(\mathbf{x}, b) \sim \mathcal{R}$. $\Phi$ denotes the Gaussian CDF.

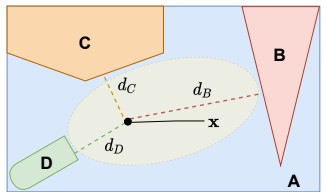

(a) If $\mathbf{x}$ is in training set

**Theorem 2 (Failure of MI)** *Given a linear classifier $f$ trained on $S$, the probability that an adversary $\mathcal{M}$ correctly predicts the membership of inputs randomly belonging to the training or test set,* $\mathbb{P}_{\mathbf{x} \sim \mathcal{R}}\left[\mathcal{M}(\mathbf{x}, f) = b\right] = 1 - \Phi\left(-\sqrt{\frac{D}{2m}}\right)$, *and decreases with* $|S| = m$. *Moreover,* $\lim_{m \to \infty} \mathbb{P}_{\mathbf{x} \sim \mathcal{R}}\left[\mathcal{M}(\mathbf{x}, f) = b\right] = 0.5$.

(b) If $\mathbf{x}$ is not in training set

The theorem suggests that the success of MI when querying a single data point is extremely low. Ass $m$ increases, the adversary can do no better than a coin flip. This means that the success is directly proportional to overfitting (we present the proof in Appendix A.3)

Figure 1: The effect of including $(\mathbf{x}, \text{'A'})$ in the train set. If $\mathbf{x}$ is in the train set, the classifier will learn to maximize the decision boundary's distance to $\mathcal{Y} \setminus \{\text{'A'}\}$. If $\mathbf{x}$ is in the test set, it has no direct impact on the learned landscape.

**Success of Dataset Inference.** Take $\mathcal{V}$ to be a dataset inferring victim (Definition 1). Let $\psi_{\mathcal{V}}(f, S; \mathcal{D})$ be $\mathcal{V}$'s decision function for ownership resolution. In the next theorem, we show that the success of DI in practice is high and independent of the training set size. (Proof in Appendix A.4)

**Theorem 3 (Success of DI)** *Choose $b \leftarrow \{0, 1\}$ uniformly at random. Given an adversary's linear classifier $f$ trained on $S' \sim \mathcal{D}$, s.t. $|S'| = |S|$ if $b = 0$, and on $S$ otherwise. The probability $\mathcal{V}$ correctly decides if an adversary stole its knowledge* $\mathbb{P}\left[\psi(f, S; \mathcal{D}) = b\right] = 1 - \Phi\left(-\frac{\sqrt{D}}{2\sqrt{2}}\right)$. *Moreover,* $\lim_{D \to \infty} \mathbb{P}\left[\psi(f, S; \mathcal{D}) = b\right] = 1$.

**Example.** Assume a dataset of training size 50K and input dimensions $K = 100, D = 900$ (i.e., 100 strongly correlated features which is roughly similar to the MNIST dataset) We have $\mathbb{P}_{(\mathbf{x}, y) \sim S}\left[\psi(f, S; \mathcal{D}) = 1\right] = 1 - 10^{-26} \sim 1.0$ while $\mathbb{P}_{(\mathbf{x}, b) \sim \mathcal{R}}\left[\mathcal{M}(\mathbf{x}, f) = b\right] = 0.526$. Therefore, in a problem setting where membership inference succeeds only by slightly above random chance, dataset inference succeeds nearly every time.

## 5 DATASET INFERENCE

Dataset Inference is the process of determining whether a victim's private knowledge has been directly or indirectly incorporated in a model trained by an adversary. Our key intuition is that classifiers generally try to maximize the distance of training examples from the model's decision boundaries. This means that any model which has stolen the victim's private knowledge should also position data similar to victim's private training data far from its own decision boundaries. (See Figure 1) When a victim suspects knowledge was stolen from their model, they may measure how the adversary's model responds to their own training data to substantiate their ownership claim.

### 5.1 EMBEDDING GENERATION

For a model $f$ and data point $\mathbf{x}$, we aim to extract a feature embedding for $\mathbf{x}$ that characterizes its 'prediction margin' (or distance from the decision boundaries) w.r.t. $f$. The victim $\mathcal{V}$ extracts these embeddings for points $(\mathbf{x}, y) \sim \mathcal{D}$ and labels them as inside $(b = 1)$ or outside $(b = 0)$ of their private dataset $S_{\mathcal{V}}$.[2] We introduce two methods for generating embeddings based on the level of access the victim may have to the adversary's model.

---

[2]Recall that for our discussion on linear networks in § 4, we used a simple metric to compute the 'prediction margin' of a given data point as $(y \cdot f(\mathbf{x}))$. However, the same does not apply to deep networks.

**White-Box Setting:** *MinGD* White-box embedding generation is used when $\mathcal{V}$ and $\mathcal{A}_*$ resolve the claim for ownership in the presence of a neutral arbitrator, such as a court. Indeed, Kumar et al. (2020) highlight that such attacks potentially fall under Computer Fraud and Abuse Act in the USA and are prosecutable for 'reverse engineering' the model's 'source code'. Both parties provide access to their models, and then the 'prediction margin' is measured for the suspected adversary's model on the victim's train and test data points. For any data point $(\mathbf{x}, y)$ we evaluate its minimum distance $\Delta$ to the neighbouring target classes $t$ by performing gradient descent optimization of the following objective (Szegedy et al., 2013): $\min_\delta \Delta(\mathbf{x}, \mathbf{x} + \delta)$ *s.t.* $f(\mathbf{x} + \delta) = t$. The distance metric $\Delta(\mathbf{x}^{(i)}, \mathbf{x}^{(j)})$ refers to the $\ell_p$ distance between points $\mathbf{x}^{(i)}$ and $\mathbf{x}^{(j)}$ for $p \in \{1, 2, \infty\}$, and $t$ is the target label. The distance $\Delta$ to each target class is a feature in the embedding vector analyzed by the ownership tester from §5.2.

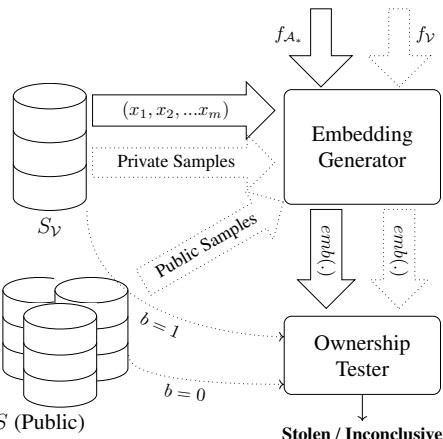

Figure 2: Training (dotted) the confidence regressor with embeddings of public and private data, and victim's model $f_\mathcal{V}$; Dataset Inference (solid) using $m$ private samples and adversary model $f_{\mathcal{A}_*}$

**Black-Box Setting:** *Blind Walk.* $\mathcal{V}$ may want to perform DI on a publicly deployed model $f$ that only allows label query access. This makes them incapable of computing gradients required for *MinGD*. Moreover, querying $f$ would be costly for $\mathcal{V}$. Therefore, we introduce a new method called *Blind Walk* which estimates the 'prediction margin' of any given data point through its robustness to random noise rather than a gradient search. We sample a random initial direction $\delta$. Starting from an input $(\mathbf{x}, y)$, we take $k \in \mathbb{N}$ steps in the same direction until $f(\mathbf{x} + k\delta) = t$; $t \neq y$. Then, $\Delta(\mathbf{x}, \mathbf{x} + k\delta)$ is used as a proxy for the 'prediction margin' of the model. Thus, the approach only requires label access to $f$. We repeat the search over multiple random initial directions to increase the information about the point's robustness, and use each of these distance values as features in the generated embedding. In practice, we find *Blind Walk* to perform better than *MinGD* with the ownership tester from §5.2. We discuss further details justifying these observations in Appendix C.

## 5.2 OWNERSHIP TESTER

It is important for the victim to resolve ownership claims in as few queries as possible, since each query involves the victim revealing part of their private dataset $S_\mathcal{V}$. Since claiming ownership would likely lead to legal action, it is paramount that the victim minimizes their false positive rate. We thus test ownership in two phases: a regression model first infers whether the potentially stolen model's predictions on individual examples contain the victim's private knowledge, this is then followed by a hypothesis test which aggregates these results to decide dataset inference. This is another key difference with membership inference efforts: rather than always predicting that a point is from the 'train' or 'test' data, we claim ownership of a model only when we have sufficient confidence. This is done through statistical hypothesis testing, which takes the false positive rate $\alpha$ as a hyper-parameter, and produces either conclusive positive results with an error of at most $\alpha$, or an 'inconclusive' result.

**Confidence Regressor.** As defined in § 5.1, we extract distance embeddings w.r.t $f_\mathcal{V}$ for data points in both $\mathcal{V}$'s private data $S_\mathcal{V}$ and unseen publicly available data. Using the embeddings and the ground truth membership labels, we train a regression model $g_\mathcal{V}$. The goal of $g_\mathcal{V}$ is to predict a (proxy) measure of confidence that a sample contains $f_\mathcal{V}$'s private information. For our hypothesis testing, we require that $g_\mathcal{V}$ produce smaller values for samples from $S_\mathcal{V}$. Complete access to the dataset $S_\mathcal{V}$ allows $\mathcal{V}$ to train $g_\mathcal{V}$ accurately, as illustrated via dotted arrows in Figure 2.

**Hypothesis Testing.** This is the step where dataset inference claims are made (solid lines in Figure 2). Using the confidence scores produced by $g_\mathcal{V}$ and the membership labels, we create equal-sized sample vectors $\mathbf{c}$ and $\mathbf{c}_\mathcal{V}$ from private training and public data, respectively. We test the null hypothesis $H_0 : \mu < \mu_\mathcal{V}$ where $\mu = \bar{\mathbf{c}}$ and $\mu_\mathcal{V} = \bar{\mathbf{c}}_\mathcal{V}$ are mean confidence scores. The test would either reject $H_0$ and conclusively rule that $f_{\mathcal{A}_*}$ is 'stolen', or give an inconclusive result.

# 6 Experimental Setup and Implementation of Dataset Inference

Unlike prior work on membership inference, which evaluates over victim models trained to overfit on small subsets of the original dataset, we train all of our victim models on large common benchmarks.

**Datasets.** We perform our experiments on the CIFAR10, CIFAR100, SVHN and ImageNet datasets. These remain popular image classification benchmarks, further description about which can be found in Appendix E.1. All details about experiments on SVHN and ImageNet are in Appendix E.2.

**Model Architecture.** The victim model is a WideResNet (Zagoruyko & Komodakis, 2016) with depth 28 and widening factor of 10 (WRN-28-10) for both CIFAR10 and CIFAR-100, and is trained with a dropout rate of 0.3 (Srivastava et al., 2014). For the model stealing attacks described in § 6.1, we use smaller architectures such as WRN-16-1 on CIFAR10 and WRN-16-10 on CIFAR100.

## 6.1 Model Stealing Attacks

We consider the strongest model stealing attacks in the literature, and introduce new attacks targeting dataset inference to perform an adaptive evaluation of our defense. The adversary $\mathcal{A}_*$ can gain different levels of access to $\mathcal{V}$'s private knowledge:

**(1)** $\mathcal{A}_Q$ has query access to $f_\mathcal{V}$. We consider model extraction (Tramèr et al., 2016) based adversaries which may **(1.a)** have access to the model's prediction vectors (via an API). $\mathcal{A}_Q$ queries $f_\mathcal{V}$ on a non-task specific dataset, and minimizes the $KL$ divergence with its predictions. **(1.b)** Alternately, to further distance its predictions from the victim, the adversary may only use the most confident label from these queries (as pseudo-labels) to train. **(2)** $\mathcal{A}_M$ has access to the victim's model $f_\mathcal{V}$. This may happen when $\mathcal{V}$ wishes to open-source their work for academic purposes but does not allow its commercialization, or via insider-access. **(2.a)** $\mathcal{A}_M$ may fine-tune over $f_\mathcal{V}$, or **(2.b)** use $f_\mathcal{V}$ for data-free distillation (Fang et al., 2019) in a zero-shot learning framework that only utilizes synthetic and non-semantic queries.[3] **(3)** $\mathcal{A}_D$ has access to the complete private dataset, $S_\mathcal{V}$ of the victim. They may train their own model either **(3.a)** by distilling $f_\mathcal{V}$ (over query access), or **(3.b)** training from scratch using different learning schemes or architectures. (For further details see Appendix B).

Finally, we also perform DI against an independent and honest machine learning model $\mathcal{I}$ that was trained on its own private dataset. This model is used as a control, to ensure that we do not claim ownership of models that were not trained by stealing knowledge from our victim model.

**Training the threat models.** For model extraction and fine-tuning attacks on CIFAR10 and CIFAR100, we use a subset of 500,000 unlabeled TinyImages that are closest to CIFAR10, as created by Carmon et al. (2019). For SVHN, we use the 'extra' training data released by the authors. We train the student model for 20 epochs for model extraction methods and 5 epochs for fine-tuning. For zero-shot learning, we use data-free adversarial distillation method (Fang et al., 2019) and train the student model for 200 epochs. In case of distillation and modified architecture, we have access to the original training data of the victim. We train both models for 100 epochs on the full training set.

In all the training methods, we use a fixed learning rate strategy with SGD optimizer and decay the learning rate by a factor of 0.2 at the end of the $0.3\times$, $0.6\times$, and $0.8\times$ the total number of epochs.

## 6.2 Implementation Details for Dataset Inference

**Embedding generation.** For the white-box method (MinGD), we perform the attack against each target class while optimizing the $\ell_1, \ell_2, \ell_\infty$ norms. Hence, we obtain an embedding of size 30 (classes$\times$distance measures). In the case of CIFAR100, we only attack the 10 most confident target classes, as indicated by the prediction vector $f(\mathbf{x})$. For the black-box method (*Blind Walk*), we sample 10 times from uniform, Gaussian, and laplace distributions to perturb the input. Once again, we obtain an embedding vector of size 30. More details are deferred to Appendix C.

**Training the confidence regressor.** We train a two-layer linear network (with tanh activation) $g_\mathcal{V}$ for the task of providing confidence about a given data point's membership in 'private' and 'public' data. The regressor's loss function is $\mathcal{L}(\mathbf{x}, y) = -b \cdot g_\mathcal{V}(\boldsymbol{x})$ where the label $b = 1$ for a point in the (public) training set of the respective model, and $-1$ if it came from victim's private set.

---

[3]This is the first work to consider data-free distillation as a stealing attack.

**Hypothesis Tests.** We query models with equal number of samples from public and private datasets, create embeddings and calculate confidence score vectors $c$ and $c_\mathcal{V}$, respectively. We form a two sample T-test on the distribution of $c$ and $c_\mathcal{V}$ and calculate the p-value for the one-sided hypothesis $\mathcal{H}_0 : \mu < \mu_\mathcal{V}$ against $\mathcal{H}_{alt} : \mu > \mu_\mathcal{V}$. From $\mathcal{L}(\mathbf{x}, y)$, it follows that $g_\mathcal{V}$ learns to minimize $g_\mathcal{V}(\mathbf{x})$ when $\mathbf{x} \in S_\mathcal{V}$, and maximizes it otherwise. Therefore, a vector that contains samples from $S_\mathcal{V}$ produces lower confidence scores, and decreases the test's p-value. If the p-value is below a predefined significance level $\alpha$, $H_0$ is rejected, and the model under test is marked as 'stolen'.

In the following results, we repeat all experimental statistical tests for 100 times with randomly sampled data with replacement. To control for multiple testing, and account for the unknown dependence of the p-values thus generated, we aggregate these values using the harmonic mean (Wilson, 2018). To produce bootstrap 99-percentile confidence intervals, we repeat the experiment 40 times.

## 7 RESULTS

Table 1 shows p-values and the effect size, $\Delta\mu = \mu - \mu_\mathcal{V}$, which captures the average confidence of our hypothesis test in claiming that a model was stolen. We test our approach against 6 different attackers and in two different settings (Black- and White-box). In addition, Table 1 also reports 'Source' where the victim's complete model $f_\mathcal{V}$ has been stolen, and 'Independent', the control model trained on a separate dataset. Understandably, we typically observe the largest and smallest effect sizes for these two baselines, which serve as bounds to interpret our evaluation of attacks.

Our evaluation shows that DI is robust to both the strongest model stealing techniques, but also an adaptive attack we propose based on zero-shot learning. DI can claim a model was stolen with at least 95% confidence for most threat models with only 10 samples. Hence, the defense exploits an inherent property of model training. Among the six attacks we considered, we observe that our model consistently flags fine-tuned models as stolen. This departs from prior defenses against model extraction: e.g., watermarks often lack robustness to fine-tuning. Here, DI is unaffected because fine-tuning does not remove knowledge from all private data used to train the stolen model. The label-query and zero-shot attacks challenge DI the most. This is expected because zero-shot learning uses only synthetic data points for querying; and in case of logit-query, $\mathcal{A}_Q$ is merely using $\mathcal{V}$ to label their dataset, which leaks much less private knowledge than distillation-based model extraction. In practice, their higher query complexity makes both these attacks the most (financially) expensive to mount. We present concurring results on SVHN and ImageNet in Appendix E.2.

**DI requires few private points.** In Figure 3, we show the number of private points the victim has to reveal (from its training set) to achieve a particular p-value when claiming model ownership is low: 40, and often as few as 20, samples to achieve a false positive rate (FPR) $\alpha$ of at most 1%.

**Query efficiency.** For the black-box scenario where the victim wants to assess the ownership of a model served through an API, DI is a query efficient approach that comes at a low cost for the victim.

| | Model Stealing Attack | CIFAR10 | | | | CIFAR100 | | | |
|---|---|---|---|---|---|---|---|---|---|
| | | MinGD | | Blind Walk | | MinGD | | Blind Walk | |
| | | $\Delta\mu$ | p-value | $\Delta\mu$ | p-value | $\Delta\mu$ | p-value | $\Delta\mu$ | p-value |
| $\mathcal{V}$ | Source | 0.838 | $10^{-4}$ | 1.823 | $10^{-42}$ | 1.219 | $10^{-16}$ | 1.967 | $10^{-44}$ |
| $\mathcal{A}_D$ | Distillation | 0.586 | $10^{-4}$ | 0.778 | $10^{-5}$ | 0.362 | $10^{-2}$ | 1.098 | $10^{-5}$ |
| | Diff. Architecture | 0.645 | $10^{-4}$ | 1.400 | $10^{-10}$ | 1.016 | $10^{-6}$ | **1.471** | $10^{-14}$ |
| $\mathcal{A}_M$ | Zero-Shot Learning | **0.371** | $10^{-2}$ | **0.406** | $10^{-2}$ | 0.466 | $10^{-2}$ | 0.405 | $10^{-2}$ |
| | Fine-tuning | **0.832** | $10^{-5}$ | **1.839** | $10^{-27}$ | **1.047** | $10^{-7}$ | 1.423 | $10^{-10}$ |
| $\mathcal{A}_Q$ | Label-query | 0.475 | $10^{-3}$ | 1.006 | $10^{-4}$ | **0.270** | $10^{-2}$ | **0.107** | $10^{-1}$ |
| | Logit-query | 0.563 | $10^{-3}$ | 1.048 | $10^{-4}$ | 0.385 | $10^{-2}$ | 0.184 | $10^{-1}$ |
| $\mathcal{I}$ | Independent | 0.103 | 1 | -0.397 | 0.675 | -0.242 | 0.545 | -1.793 | 1 |

Table 1: Ownership Tester's effect size (higher is better) and p-value (lower is better) using $m = 10$ samples on multiple threat models (see § 6.1). The highest and lowest effect sizes among the model stealing attacks ($\mathcal{A}_D, \mathcal{A}_M, \mathcal{A}_Q$) are marked in **red** and **blue** respectively.

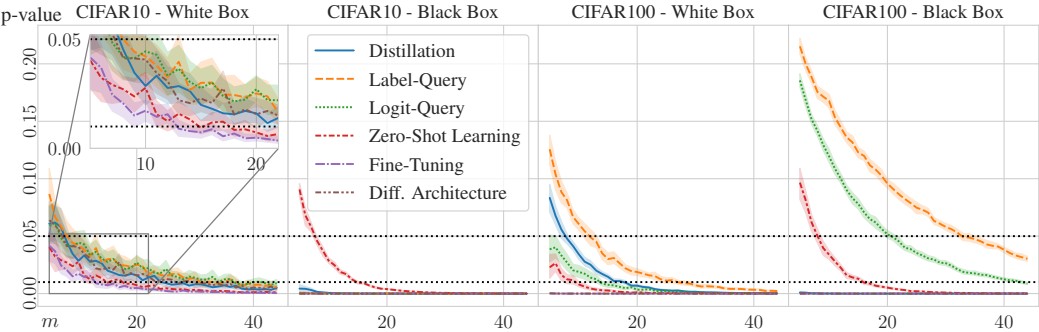

Figure 3: p-value against number of revealed samples ($m$). Significance levels (FPR) $\alpha = 0.01$ and 0.05 (dotted lines) have been drawn. Under most attack scenarios, the victim $\mathcal{V}$ can dispute the adversary's ownership of $f_{\mathcal{A}_*}$ (with FPR of at most 1%) by revealing fewer than 50 private samples.

For 100 data points, DI can be performed in less than 30,000 queries to the API. More efficient embedding generation optimizations can significantly improve this further. (See Appendix D).

**White-box access is not essential to DI.** While access to gradient information can help in particular scenarios (such as, logit-query in CIFAR100) which reduces $m$ from 40 to 20 samples, for fine-tuned adversary models, or those that are trained against a different architecture to evade detection, our proposed black-box solution (Blind Walk) performs surprisingly better than its White-box counterpart. We conjecture that the Blind Walk's advantage stems from a combination of factors: (a) gradient-based approaches are sensitive to numerical instabilities, (b) the approach is stochastic and aims to find the expected prediction margin rather than the worst-case (it searches for *any* incorrect neighboring class in a randomly chosen direction rather than focusing on the distance to possible *target* classes). Hence, **our proposed *Blind Walk* inference procedure is highly efficient**.

**DI does not require overfitting or retraining.** Unlike past defenses (watermarks) and attacks (MI) which we discussed previously, DI uniquely applies as a post-hoc solution to any publicly deployed model, irrespective of whether it 'overfit' on its training set. This means that model owners in the real-world can perform DI immediately, to protect models that they have already deployed.

## 8 DISCUSSION AND CONCLUSION

While adversarial ML often consists of a cycle of attacks and defenses, we turn this game on its head. Dataset inference leverages knowledge a defender has of their training set to identify models that an adversary created by either directly accessing this training set without authorization or indirectly distilling knowledge from one of the models released by the defender. With dataset inference, model developers resolve model ownership conflicts without making changes to their existing models.

Interestingly, the ability to claim ownership through dataset inference gracefully degrades as the adversary spends increasingly more resources to train the stolen model. For instance, if an adversary extracts a copy and later fine-tunes it with a different dataset to conceal the model, it will make the model more different and *dataset inference* will be less likely to succeed. But this is expected and desired: this means the adversary faced a higher cost to obfuscate this stolen copy. In itself it is not an easy task, because of accuracy degradation and catastrophic forgetting.

Finally, it remains a promising direction for future work to study the confluence of DI with privacy-preserving models trained using $\epsilon$-differential privacy (DP). Leino & Fredrikson (2020) have shown that while DP can help against membership inference (MI) attacks, it comes at a steep cost in accuracy. We hypothesize that since DI amplifies the membership signal using multiple *private* samples, it follows that the $\epsilon$ values required to make DI ineffective would be even lower than it is for MI. Therefore, $\epsilon$ values that can make the model private, likely do not interfere with dataset inference.

**Acknowledgments.** We thank the reviewers for their feedback. We would also like to thank members of CleverHans Lab, especially Ilia Shumailov and Varun Chandrasekaran. This work was supported by a Canada CIFAR AI Chair, NSERC, Microsoft, and sponsors of the Vector Institute.

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

APPENDIX

# A  THEORETICAL MOTIVATION

In this section, we provide the formal proofs of Theorems 1, 2, 3 as stated in § 4. First, we describe the preliminaries including the binary classification task and the machine learning model used to train the same in Appendix A.1.

## A.1  PRELIMINARIES

We repeat the preliminaries described in § 4 to discuss the proofs in the following sections.

**Setup.**  Consider a data distribution $\mathcal{D}$, such that any input-label pair $(\mathbf{x}, \mathbf{y})$ can be described as:

$$y \overset{u.a.r}{\sim} \{-1, +1\}; \quad \mathbf{x_1} = y\mathbf{u} \in \mathbb{R}^K, \quad \mathbf{x_2} \sim \mathcal{N}(0, \sigma^2) \in \mathbb{R}^D \tag{2}$$

where $\mathbf{x} = (\mathbf{x_1}, \mathbf{x_2}) \in \mathbb{R}^{K+D}$ and $\mathbf{u} \in \mathbb{R}^K$ is some fixed vector. This suggests that the last $D$ dimensions of the input is Gaussian noise which has no correlation with the correct label. However, the first $K$ input dimensions are sufficient to perfectly separate data points from classes $\{-1, +1\}$. The setup is adapted from Nagarajan & Kolter (2019). We use $S^+$ and $\mathcal{D}^+$ to represent the subset of the training set $S$ and the distribution $\mathcal{D}$ with label $y = 1$.

**Architecture.**  We consider the scenario of classifying the input distribution using a linear classifier, $f$, with weights $\mathbf{w} = (\mathbf{w_1}, \mathbf{w_2})$, such that for any input:

$$f(\mathbf{x}) = \mathbf{w_1} \cdot \mathbf{x_1} + \mathbf{w_2} \cdot \mathbf{x_2} \tag{3}$$

While we only discuss the case of a linear network in this analysis, the success of dataset inference (like membership inference) only increases with the number of parameters in a machine learning model (Yeom et al., 2018), which in effect makes the following analysis a stronger result to prove. Prior works have also argued how over-parametrized deep learning networks memorize training points (Zhang et al., 2016; Feldman, 2019).

## A.2  TRAIN-TEST PREDICTION MARGIN (THEOREM 1)

**Training Algorithm.**  We assume that the learning algorithm initializes the weights of the classifier $f$ to zero. Sample a training set $S \sim \mathcal{D}^m = \left\{ \left( \mathbf{x}^{(i)}, y^{(i)} \right) \mid i = 1 \ldots m \right\}$. The learning algorithm maximizes the loss $\mathcal{L}(\mathbf{x}, y) = y \cdot f(\mathbf{x})$ and visits every training point once, with a gradient update step of learning rate $\alpha = 1$.

$$\begin{aligned} \mathbf{w_1} &\leftarrow \mathbf{w_1} + \alpha y^{(i)} \mathbf{x_1}^{(i)} \\ \mathbf{w_2} &\leftarrow \mathbf{w_2} + \alpha y^{(i)} \mathbf{x_2}^{(i)} \end{aligned} \tag{4}$$

From the optimization steps described above, one may note that the learned weights for the classifier $f$ are given by $\mathbf{w_1} = m\mathbf{u}$ and $\mathbf{w_2} = \sum_i y^{(i)} \mathbf{x_2}^{(i)}$ irrespective of the training batch size.

**Inference.**  For any data point $(\mathbf{x}^{(j)}, y^{(j)})$, we calculate its 'prediction margin' as the distance from the linear boundary, which is proportional to its label times the classifier's output $y \cdot f(\mathbf{x})$. For any point, $\mathbf{x} = (x_1, x_2) \sim \mathcal{D}$, the 'prediction margin' is therefore given by:

$$\begin{aligned} y \cdot f(\mathbf{x}) &= y \cdot (\mathbf{w_1} \cdot \mathbf{x_1} + \mathbf{w_2} \cdot \mathbf{x_2}) = y \cdot (m\mathbf{u}) \cdot (y\mathbf{u}) + y \cdot \left( \sum_i y^{(i)} \mathbf{x_2}^{(i)} \right) \cdot \mathbf{x_2} \\ &= c + \left( y \cdot \sum_i y^{(i)} \mathbf{x_2}^{(i)} \cdot \mathbf{x_2} \right) \end{aligned} \tag{5}$$

Now, we calculate the expected value of the margin for a point randomly sampled from the training set. Consider any point in the training set $(\mathbf{x}, y) \sim S^+ = (\mathbf{x}^{(j)}, 1)$ for some index $j$. Then, we have:

$$\mathbb{E}_{\mathbf{x}^{(j)} \sim S^+} f(\mathbf{x}^{(j)}) = y \cdot c + \mathbb{E}_{\mathbf{x_2}^{(i)} \sim \mathcal{N}(0,\sigma^2)} \left[ \left( \sum_i^{i \neq j} y^{(i)} \mathbf{x_2}^{(i)} \cdot \mathbf{x_2}^{(j)} \right) \right] + \mathbb{E}_{\mathbf{x_2}^{(j)} \sim \mathcal{N}(0,\sigma^2)} \left[ y^{(j)} (\mathbf{x_2}^{(j)})^2 \right]$$

$$= c + 0 + D\sigma^2$$

(6)

Note that in (6), we utilize the fact that the square of a standard normal variable follows the $\chi^2_{(1)}$ distribution; and that the expected value of product of independent random variables is same as the product of their expectations, followed by the linearity of expectation.

Similarly, now consider a new data point $(\mathbf{x}, 1) \sim \mathcal{D}^+$.

$$\mathbb{E}_{(\mathbf{x},y) \sim \mathcal{D}^+} f(\mathbf{x}) = yc + \mathbb{E}_{\mathbf{x_2}^{(i)} \sim \mathcal{N}(0,\sigma^2)} \left[ \left( \sum_i y^{(i)} \mathbf{x_2}^{(i)} \cdot \mathbf{x_2} \right) \right]$$

$$= c$$

(7)

Once again, in (7) we utilize the fact that the expected value of product of independent random variables is same as the product of their expectations, followed by the linearity of expectation. At an aggregate over multiple data points, we can hence show that $\mathbb{E}_{(\mathbf{x},y) \sim S^+} f(\mathbf{x}) - \mathbb{E}_{(\mathbf{x},y) \sim \mathcal{D}^+} f(\mathbf{x}) = D\sigma^2$. This concludes the proof for Theorem 1.

## A.3 FAILURE OF MEMBERSHIP INFERENCE (THEOREM 2)

In this section, we take a formal view of the conditions that lead to the failure and success of membership inference. Before we begin with our formal analysis, we would like to point out that the statistical difference between the distribution of training and test data points in Theorem 1 is only observed when we aggregate an expectation over multiple samples. Now, we show that the variance of this difference is so large, that it is very difficult to make any claims from a single input data point.

Consider an adversary that does not have knowledge of the private data used to train a machine learning model. However, it contains domain knowledge of the task that the model is trying to solve. This may include the range and dimension of possible inputs to the model. In our case, the adversary has knowledge of the data distribution $\mathcal{D}$, but not of the training set $S$.

For a single data point $\mathbf{x} = (\mathbf{x_1}, \mathbf{x_2})$, $s.t.$ $(\mathbf{x}, y) \sim \mathcal{D}$, the adversary aims to predict whether it was used to train the machine learning model, $f$. The prediction margin for $(\mathbf{x}, y) \sim \mathcal{D}$ is given by:

$$y \cdot f(\mathbf{x}) = y \cdot (\mathbf{w_1} \cdot \mathbf{x_1} + \mathbf{w_2} \cdot \mathbf{x_2}) = c + \left( y \cdot \sum_i y^{(i)} x_2^{(i)} \cdot \mathbf{x_2} \right)$$

(8)

From the analysis in Theorem 2, the adversary knows that $\mathbb{E}_{(\mathbf{x},y) \sim S} [y \cdot f(\mathbf{x})] = c + D\sigma^2$ and $\mathbb{E}_{(\mathbf{x},y) \sim \mathcal{D}} [y \cdot f(\mathbf{x})] = c$. Let $\mathcal{M}(\mathbf{x}|f)$ represents the membership decision of the adversary for a given data point $\mathbf{x}$ and classifier $f$. The strongest adversary will use the following decision rule:

$$\mathcal{M}(\mathbf{x}|f) = \begin{cases} 1, & \text{if } (y \cdot f(\mathbf{x}) - c) \geq t \\ 0, & o.w. \end{cases}$$

(9)

where $t \in [0, D\sigma^2]$ is some threshold that the adversary can tune in order to achieve maximum true positive rate and minimum false positives.

Similar to Yeom et al. (2018), we consider the scenario where the input data is randomly (with equal probability via coin flip $b$) sampled from either $S$ (if $b = 1$) or $\mathcal{D}$ (if $b = 0$). Let such a distribution be specified as $(\mathbf{x}, b) \sim \mathcal{R}$. The adversary $\mathcal{M}$ must maximize the single objective $\mathbb{P}_{(\mathbf{x},b) \sim \mathcal{R}} [\mathcal{M}(\mathbf{x}|f) = b]$. In summary,

$$\mathbb{P}_{(\mathbf{x},b) \sim \mathcal{R}} [\mathcal{M}(\mathbf{x}|f) = b] = \frac{\mathbb{P}_{(\mathbf{x},y) \sim S} [\mathcal{M}(\mathbf{x}|f) = 1] + \mathbb{P}_{(\mathbf{x},y) \sim \mathcal{D}} [\mathcal{M}(\mathbf{x}|f) = 0]}{2}$$

(10)

We simplify our analysis by considering the data point $(\mathbf{x}, y) \sim \mathcal{D}^+$ (has true label, $y = 1$). However, the analysis generally applies to any $(\mathbf{x}, y) \sim \mathcal{D}$.

**Case 1:** $(\mathbf{x}, y) \sim \mathcal{D}^+$. Assume meta-variable $\mathbf{z_2} = \left(\sum_i y^{(i)} \mathbf{x_2}^{(i)}\right)$. Therefore, $\mathbf{z_2} \sim \mathcal{N}(0, m\sigma^2 I)$, while $\mathbf{x_2} \sim \mathcal{N}(0, \sigma^2 I)$. Recall that $\mathbf{x_2}, \mathbf{z_2} \in \mathbb{R}^D$. Assuming $D$ to be large, we can conveniently apply the central limit theorem to approximate the distribution of the internal term. Let the individual dimensions of $\mathbf{z_2}$ be denoted by $\mathbf{z_2}(i)$. Then, we have that:

$$
\begin{aligned}
\mathbb{P}_{(\mathbf{x},y)\sim\mathcal{D}^+}[\mathcal{M}(\mathbf{x}|f) = 0] &= \mathbb{P}_{(\mathbf{x},y)\sim\mathcal{D}^+}\left[\left(\sum_i^m y^{(i)} \mathbf{x_2}^{(i)} \cdot \mathbf{x_2}\right) < t\right] \\
&= \mathbb{P}_{(\mathbf{x},y)\sim\mathcal{D}^+}\left[(\mathbf{z_2} \cdot \mathbf{x_2}) < t\right] \\
&= \mathbb{P}_{(\mathbf{x},y)\sim\mathcal{D}^+}\left[\frac{1}{D}\left(\sum_j^D D\mathbf{z_2}(j) \cdot \mathbf{x_2}(j)\right) < t\right]
\end{aligned}
\tag{11}
$$

Let $\alpha$ represents the distribution followed by $\mathbf{z_2}(i) \cdot \mathbf{x_2}(i)$. From CLT, we have that the combined distribution behaves like a normal distribution, with $\mu = \mu_\alpha =$ and $\sigma^2 = \frac{\sigma_\alpha^2}{D}$.

$$
\begin{aligned}
\mu_\alpha &= 0 \\
\sigma_\alpha^2 &= m \cdot D^2 \cdot \sigma^4
\end{aligned}
\tag{12}
$$

We use the fact that $\mathrm{Var}[XY] = \mathrm{Var}[X]\mathrm{Var}[Y] + \mathbb{E}[X]^2\mathrm{Var}[Y] + \mathbb{E}[Y]^2\mathrm{Var}[X]$ and $\mathrm{Var}[c \cdot X] = c^2 \cdot \mathrm{Var}[X]$ for computing $\sigma_\alpha^2$. Therefore, let $r \sim \mathcal{N}(0, mD\sigma^4)$:

$$
\mathbb{P}_{(\mathbf{x},y)\sim\mathcal{D}^+}[\mathcal{M}(\mathbf{x}|f) = 0] = \mathbb{P}_{r\sim\mathcal{N}(0,mD\sigma^4)}[r < t]
\tag{13}
$$

It can be observed that $\mathbb{P}[r < t]$ increases with the threshold value $t$. For $t = 0$, $\mathbb{P}[(\mathbf{z_2} \cdot \mathbf{x_2}) < 0] = 0.5$. Whereas, for $t = D\sigma^2$, the probability decreases with the value of m (this can be intuitively understood as – since the size of training set increases, overfitting decreases, making MI more difficult). Even for as low as $m = 100$ points in the training set, $\mathbb{P}[(\mathbf{z_2} \cdot \mathbf{x_2}) < D\sigma^2] = 0.6$. For any value of $t \in [0, \sigma^2]$, the maximum probability for size of training data $m = 100$ is 0.6. Further, as the size of the training set increases, the probability tends to 0.5.

**Case 2:** $(\mathbf{x}, y) \sim S^+$. Once again, as in the proof for Theorem 1, consider any point in the training set $(\mathbf{x}, y) \sim S^+ = (\mathbf{x}^{(j)}, 1)$ for some index $j$. We will now calculate the probability of success of the adversary that follows the decision rule described above:

$$
\begin{aligned}
\mathbb{P}_{(\mathbf{x},y)\sim S^+}[\mathcal{M}(\mathbf{x}|f) = 1] &= \mathbb{P}_{(\mathbf{x},y)\sim S^+}\left[\left(\sum_i y^{(i)}\mathbf{x_2}^{(i)} \cdot \mathbf{x_2}\right) > t\right] \\
&= \mathbb{P}_{(\mathbf{x},y)\sim S^+}\left[\left(\sum_i^{i\neq j} y^{(i)}\mathbf{x_2}^{(i)} \cdot \mathbf{x_2}^{(j)}\right) + \left(\mathbf{x_2}^{(j)} \cdot \mathbf{x_2}^{(j)}\right) > t\right]
\end{aligned}
\tag{14}
$$

Now, following the discussion in the first case, we know that the first term can be approximated by a variable $\alpha \sim \mathcal{N}(0, (m-1)D\sigma^4)$. Similarly, using CLT over the sum of multiple random variables sampled from a $\chi_1^2$ distribution, we can approximate the second term in the above equation with a variable $\beta \sim \mathcal{N}(D\sigma^2, D\sigma^4)$. Finally, using the property for sum of independent gaussians, we can approximate the entire 'prediction margin' to be represented by a sample $u \sim \mathcal{N}(D\sigma^2, mD\sigma^4)$. Then, we have that:

$$
\mathbb{P}_{(\mathbf{x},y)\sim S^+}[\mathcal{M}(\mathbf{x}|f) = 1] = \mathbb{P}_{u\sim\mathcal{N}(D\sigma^2,mD\sigma^4)}[u > t]
\tag{15}
$$

Hence, we show that the adversary can do no better than a coin flip. This concludes the proof for Theorem 2. The interested reader may further analyze the assertion that the optimal value of $t$ lies in $[0, D\sigma^2]$.

To resolve the optimal threshold $t$ for membership inference, we restructure the arguments as follows. Recall from (10) that the adversary aims to ensure both true positive rates and true negative

rates are high. We know:

$$\begin{aligned}
\mathbb{P}_{(\mathbf{x},y)\sim\mathcal{D}^+}\left[\mathcal{M}(\mathbf{x}|f)=0\right] &= \mathbb{P}_{r\sim\mathcal{N}(0,mD\sigma^4)}\left[r<t\right] \\
\mathbb{P}_{(\mathbf{x},y)\sim S^+}\left[\mathcal{M}(\mathbf{x}|f)=1\right] &= \mathbb{P}_{u\sim\mathcal{N}(D\sigma^2,mD\sigma^4)}\left[u>t\right] \\
\mathbb{P}_{(\mathbf{x},b)\sim\mathcal{R}}\left[\mathcal{M}(\mathbf{x}|f)=b\right] &\leq \mathbb{P}\left[u-r>0\right]
\end{aligned} \tag{16}$$

We know that both $u,r$ are sampled from normal distributions. Therefore, define $\gamma = (u - r) \sim \mathcal{N}(D\sigma^2, 2mD\sigma^4)$. This simplifies our discussion to a single normal distribution with mean $\mu_\gamma = D\sigma^2$ and variance, $\sigma_\gamma^2 = 2mD\sigma^4$. We can now calculate the CDF at $x = 0$ to evaluate the maximum probability of success of membership inference (decision taken by the optimal adversary).

Let $Z \sim \mathcal{N}(0,1)$. It can hence be shown that:

$$\begin{aligned}
\mathbb{P}[\gamma>0] = \mathbb{P}\left(\sigma_\gamma Z + \mu_\gamma\right) = \mathbb{P}\left(Z > -\frac{\mu_\gamma}{\sigma_\gamma}\right) &= 1 - \Phi\left(-\frac{\mu_\gamma}{\sigma_\gamma}\right) \\
&= 1 - \Phi\left(-\sqrt{\frac{D}{2m}}\right)
\end{aligned} \tag{17}$$

Clearly, as $m \to \infty$, $\mathbb{P}[\gamma>0] \to 0.5$. This concludes the proof for Theorem 2.

## A.4 SUCCESS OF DATASET INFERENCE (THEOREM 3)

In Theorem 2 we showed that an adversary querying a single data point can say no better than a coin flip about the presence or absence of a given data point in a model's training set. In this section, we show that when we reverse this adversarial game, the victim can utilize the information asymmetry to predict with high confidence if a potential adversary's model stole their knowledge in any form.

First, recall that the victim has access to its own private training set of size $m$. For the purposes of this proof, we call it $S_{\mathcal{V}}^m$. As the victim has complete information of the data distribution, it can randomly sample another dataset $S_0 \sim \mathcal{D}$.

The victim considers that the potential adversary's model was stolen if the mean 'prediction margin' for the points in $S_{\mathcal{V}}$ is greater than $S_0$ by some threshold parameter $\lambda$. Let $\psi_{\mathcal{V}}(f,S;\mathcal{D})$ be $\mathcal{V}$'s decision function to resolve ownership claims.

Recall that in Theorem 1 we had calculated the expected value of the difference in the prediction margin for the points in the training set versus those in the test set. In the proof of this theorem, we calculate the probability of the mean of the difference being greater than some value $\lambda$.

Now, let us calculate the probability of this margin for a data point randomly sampled from the training set. Let $t_{\mathcal{V}}$ represent the mean of the 'prediction margin' of all points in $S_{\mathcal{V}}$ for a classifier $f$. Similarly, let $t_0$ represent the mean of the 'prediction margin' of all points in $S_0$ for the classifier $f$. We will use $u_2$ to denote the last $D$ dimensions of points in $S_0$. Then,

$$\begin{aligned}
t_{\mathcal{V}} &= \frac{1}{m}\sum_j\left[\left(\sum_i^{i\neq j} y^{(i)}\mathbf{x_2}^{(i)}\cdot\mathbf{x_2}^{(j)}\right) + (\mathbf{x_2}^{(j)})^2\right] \\
&= \frac{1}{m}\sum_j(\mathbf{x_2}^{(j)})^2 + \sum_i\left[\left(\sum_i^{i\neq j} y^{(i)}\mathbf{x_2}^{(i)}\cdot\mathbf{x_2}^{(j)}\right)\right] \\
t_0 &= \frac{1}{m}\sum_j\left[\left(\sum_i^{i\neq j} y^{(i)}\mathbf{x_2}^{(i)}\cdot\mathbf{u_2}^{(j)}\right)\right]
\end{aligned} \tag{18}$$

$$\mathbb{P}\left[\psi_{\mathcal{V}}(f,S;\mathcal{D})=1\right] = \mathbb{P}\left[(t_{\mathcal{V}}-t_0)>\lambda\right]$$

Recognize the similarity of the above formulation with that discussed in the proof for Theorem 2 in Appendix A.3. Let $t = t_{\mathcal{V}} - t_0$. Then the random variable $t$ represents the a sample from the distribution of means for $\gamma$ defined in Appendix A.3. We can now directly use the Central Limit

Theorem for this proof. Therefore,

$$\mu_t = \mu_z = D\sigma^2$$
$$\sigma_t^2 = \frac{\sigma_z^2}{m} = 2D\sigma^4 \tag{19}$$

Hence, $t \sim \mathcal{N}(D\sigma^2, 2D\sigma^4)$. It is important to note that this distribution is independent of the number of training points. Hence, unlike membership inference, the success of DI is not curtailed by the lack of overfitting.

Similarly, for an honest adversary, the distribution of 'prediction margin' for points in $S_{\mathcal{V}}$ is the same as that for the points in $S_0$. It directly follows that:

$$\mathbb{P}\left[\psi_{\mathcal{V}}(f, S; \mathcal{D}) = 0\right] = \mathbb{P}\left[\hat{t} < \lambda\right] = \mathbb{P}\left[t > \lambda\right] \tag{20}$$

where, $\hat{t} \sim \mathcal{N}(0, 2D\sigma^4)$. Once again, like the proof of Theorem 2, by symmetry of two normal distributions with the same variance, and shifted means, we can find that the optimal value of the parameter $\lambda$ that maximized true positives, and minimizes false positives, $\lambda = \frac{\mu_t}{2}$.

Let $Z \sim \mathcal{N}(0, 1)$. Then it can hence be shown that:

$$\mathbb{P}[\hat{t} > \lambda] = \mathbb{P}\left(\sigma_t Z + \mu_t > \frac{\mu_t}{2}\right) = \mathbb{P}\left(Z > -\frac{\mu_t}{2\sigma_t}\right) = 1 - \Phi\left(-\frac{\mu_t}{2\sigma_t}\right)$$
$$= 1 - \Phi\left(-\frac{\sqrt{D}}{2\sqrt{2}}\right) \tag{21}$$

Clearly, as $D \to \infty$, $\mathbb{P}[\hat{t} > \lambda] \to 1.0$. This concludes the proof for Theorem 3.

## B  MODEL STEALING TECHNIQUES

In this section, we provide more details about the various threat models that we consider in this work. We also provide specific use-cases and motivation for the respective threat models, and introduce a new adaptive adversary targeted specifically against DI.

$\mathcal{V}$ **: Victim.**    The victim $\mathcal{V}$ wishes to release its machine learning model to the community, either as a service, or by open-sourcing it for non-commercial academic use. $\mathcal{V}$ wants to ensure that the deployed model is not being misused under the terms of license provided.

$\mathcal{A}_D$**: Data Access.**    The adversary $\mathcal{A}_D$ is able to gain complete access to the victim's private training data, and aims to deploy its own MLaaS by training the same. We note that labeled private training data is one of the most expensive commodities in the deployment cycle of modern machine learning systems.

1. **Model Distillation:** Traditionally, model distillation (Hinton et al., 2015) was used as a method to compress larger models by training smaller students using the logits of a teacher model. We use this as a threat model that the adversary may employ to distance its predictions from a model that was trained using hard labels from the dataset itself. The adversary requires both query access, and access to the victim's private training data for this attack.

2. **Modified Architecture:** Multiple works have attempted at identifying unique properties (or 'fingerprints') of a model by analyzing specific activations and representative features of internal model layers (Olah et al., 2017; 2018; Yin et al., 2019). We study the threat model where the adversary attempts training an alternate architecture on the victim's private dataset $\mathcal{D}_{priv}$ to valid the robustness of our method to changes in model structure.

$\mathcal{A}_M$**: Model Access.**    The use-case of such an adversary is two fold: (1) the victim open-sources their own machine learning model under a license that does not allow other individuals to monetize the same; and (2) the adversary gains insider access to the trained model of a victim. In both the cases, the adversary aims to monetize its own MLaaS and deploys their own model on the web, by modifying the original victim model to reduce the dependence on $\mathcal{K}$.

1. **Fine-tuning:** The adversary has full access to the victim's machine learning model, but not to its training data. While fine-tuning is employed used to transfer the knowledge of large pre-trained models on a given task (Devlin et al., 2018), we use it as a stealing attack, where the adversary uses the predictions of the victim model on unlabeled public data in order to modify its decision boundaries. We consider the setting where the adversary can fine-tune all layers.

2. **Zero-Shot Learning:** This is the strongest adversary that we introduce specifically targeted to evade dataset inference. To the best of our knowledge, we are the first to consider such a threat model. The adversary uses no 'direct' knowledge of the actual training data to avoid any features that it may learn as a result of the training on the victim's private data set. The adversary has complete access to the victim model, and uses data-free knowledge transfer (Micaelli & Storkey, 2019; Fang et al., 2019) to train a student model.

$\mathcal{A}_Q$**: Query Access.** Model extraction (Tramèr et al., 2016) is the most popular form of model stealing attack against deployed machine leaning models on the web. We discuss the related work on model extraction attacks in more detail in § 2. Depending on the access provided by the machine learning service, an adversary may aim to extract the model using the logits or the labels alone.

1. **Model Extraction Using Labels:** The victim model is used to provide pseudo-labels for a public dataset. The adversary trains their model on this pseudo-dataset. The key difference is that the input data points may be semantically irrelevant with respect to the task labels that the adversary's model is being trained on.

2. **Model Extraction Using Logits:** The performance of model extraction attacks can be improved when the victim provides confidence values for different output classes, rather than the correct labels itself. The adversary's model is trained to minimize the $KL$ divergence with the outputs of the victim on a public (or non-task specific) dataset.

$\mathcal{I}$ **: Independent Model.** Finally, we also study the results of dataset inference on an independent and honest machine learning model that is trained on its own private dataset. This is used as a control to verify that the dataset inference procedure does not always predict that the potential adversary stole the victim's knowledge.[4]

## C  EMBEDDING GENERATION

**Embedding Generation Hyperparameters.**

For the case of **_MinGD_** attack, we perform adversarial attacks defined by the optimization equation

$$\min_\delta \Delta(\mathbf{x}, \mathbf{x} + \delta) \ \ s.t. \ \ f(\mathbf{x} + \delta) = t; \ \ \mathbf{x} + \delta \in [0, 1]^n \tag{22}$$

The distance metric $\Delta(\mathbf{x}^{(i)}, \mathbf{x}^{(j)})$ refers to the $\ell_p$ distance between points $\mathbf{x}^{(i)}$ and $\mathbf{x}^{(j)}$ for $p \in \{1, 2, \infty\}$, and $t$ is the targeted label. To perform the optimization, we perform gradient descent with steps of size $\alpha_p$. We take a maximum of 500 steps of gradient optimization, but pre-terminate at the earliest misclassification. The step sizes for the individual perturbation types are given by $\{\alpha_\infty, \alpha_2, \alpha_1\} = \{0.001, 0.01, 0.1\}$.

For the case of **_Blind Walk_**, We sample a random initial direction $\delta$. Starting from an input $(\mathbf{x}, y)$, take $k \in \mathbb{N}$ steps in the same direction until $f(\mathbf{x} + \delta) = t; \ t \neq y$. Then, $\Delta(\mathbf{x}, \mathbf{x} + k\delta)$ is used as a proxy for the 'prediction margin' of the model. We repeat the search over multiple random initial directions to increase the information about a training data point's robustness, and use each of these distance values as features in the generated embedding.

As an implementation detail, we sample between uniform, laplace and gaussian noise to generate embedding features. To measure the final perturbation distance from the initial starting point, we use different $\ell_p$ norms for each of the noise sampling methods. For uniform noise, we compute the $\ell_\infty$

---

[4]Note that since we consider the difference in the distribution of outputs of the auxiliary classifier on embeddings from the test and training set (rather than hard labels from the auxiliary classifier), even in the absence of this control, we can un-deniably verify the confidence of dataset inference. This is only included to contrast the difference and make the effects of the method clearer to the reader.

distance; for gaussian noise, the $\ell_2$ distance; and for laplacian noise, the $\ell_1$ distance of the nearest misclassification. While we take $k$ steps of *Blind Walk* up till misclassification, however, we do not exceed more than 50 steps and prematurely terminate without misclassification in the event that the prediction label does not change.

**Performance of White Box Approach.**  We find in our evaluations that the white-box **MinGD** method generally underperforms the *Blind Walk* method. This happens despite its ability of being able to compute the nearest distance to any target class more accurately. While on the onset, this may seem to be a counter-intuitive result, since generally with more access, the performance of mapping the neighbours should only increase.

However, we note an important distinction. The end goal of the query generation process is not to calculate the minimum distance to target classes accurately, but rather to understand the 'prediction margin' or the local landscape of a given data point. Readers may recall from adversarial examples literature (Szegedy et al., 2013) that adversarial examples can easily be constructed on the dataset that a given machine learning model was trained on. This observation hurts the idea of Figure 1b. Despite pushing the neighbouring class boundaries away, the existence of adversarial examples elucidates the existence of small pits within the landscape of the model.

We hypothesize that the gradient-based optimization objective aims to capture this minimum (adversarial) distance, and fails to capture a 'prediction margin' that is more representative of the classifier's prediction confidence or general landscape. On the contrary, *Blind Walk* is able to perform a spectacular job at the same end goal. Since we are no longer adversarially trying to optimize the minimum distance to the neighbouring classes, multiple *Blind Walk* runs effectively map the **'average case'** prediction margin, which we argue is more useful than the **'worst case'** prediction margin as obtained by **MinGD**.

## D   EFFECT OF EMBEDDING SIZE

For all models, richer embeddings reduce the need for more revealed samples. (See Figure 4). We note that in the main body of this work, we had used a fixed size of embedding vector, with 30 input features. However, recall that in the black-box setting, the victim incurs additional cost for querying the potential adversary. Therefore, in this section we aim to understand the marginal utility of extra embedding features added. In general, we find that for most of the threat models studied, using only 10 features for the embedding space is sufficient to achieve the required threshold p-value of $0.01$. This suggests that we can slash the number of queries made to the potential adversary by one-thirds, without loss in confidence of prediction.

Interestingly, we also note that even in scenarios where the victim reveals only 15 samples, additional embedding features have insignificant advantage as opposed to querying fresh samples. This suggests that the amount of entropy gained by revealing a new data point is significantly more than that by extracting more features (beyond 10) for the same data point. We also note that the effect is not-consistent in the zero-shot learning threat model.

## E   EXPERIMENTS

### E.1   DATASET DESCRIPTION

**CIFAR10.**  CIFAR10 (Krizhevsky, 2012) contains 60,000 coloured images with 10,000 reserved for testing. There are 10 target classes with 5000 training images per class.

**SVHN.**  SVHN (Netzer et al., 2011) is a dataset obtained from house numbers in Google Street View images. The underlying task is of digit classification from $32 \times 32$ coloured images.

**CIFAR100.**  CIFAR100 (Krizhevsky, 2012) also contains 60,000 coloured images with 10,000 reserved for testing. There are 100 target classes with 500 training images per class.

**ImageNet.**  The ImageNet dataset (Deng et al., 2009) is a large-scale benchmark, consisting various challenges including that of image recognition for machine learning systems.

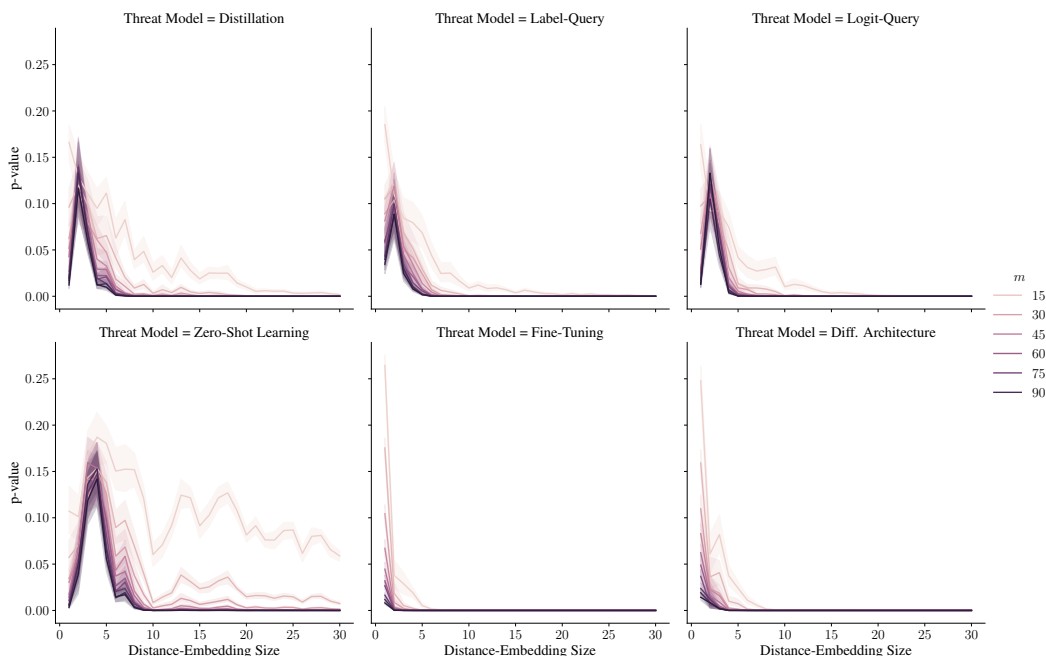

Figure 4: p-value vs. distance-embedding size

| Dataset Inference on SVHN (Blind Walk Attack) | | | |
|---|---|---|---|
| Model Stealing Attack | | $\Delta\mu$ | p-value |
| $\mathcal{V}$ | Source | 0.950 | $10^{-8}$ |
| $\mathcal{A}_D$ | Distillation | 0.537 | $10^{-3}$ |
| | Diff. Architecture | **0.450** | $10^{-2}$ |
| $\mathcal{A}_M$ | Zero-Shot Learning | 0.512 | $10^{-3}$ |
| | Fine-tuning | **0.581** | $10^{-4}$ |
| $\mathcal{A}_Q$ | Label-query | 0.513 | $10^{-03}$ |
| | Logit-query | 0.515 | $10^{-02}$ |
| | Random-query | 0.475 | $10^{-02}$ |
| $\mathcal{I}$ | Independent | -0.322 | $10^{-01}$ |

Table 2: Ownership Tester's effect size in a small-data regime (using only $m = 10$ samples) on the SVHN dataset using Blind Walk attack. 2nd highest and lowest effect size is marked in **red** & **blue**.

## E.2 ADDITIONAL DATASETS

To further validate our claims about the success of *dataset inference*, we provide evidence on two additional datasets. In this section, we present results on the SVHN and ImageNet datasets.

**SVHN.** The results of DI via the 'Blind Walk' attack on the various threat models discussed in Appendix B are presented in Table 2. To perform model extraction and fine-tuning attacks, we utilized the set of 'extra' images available with the SVHN dataset. We use the first 50,000 images in this set to stage such attacks that require a surrogate dataset. For training on the original dataset with a different architecture, we once again utilize the Pre-activation version of ResNet-18 as for CIFAR10 and CIFAR100 in the main paper. Notably, we also introduce another threat model: *Random-query* which describes a scenario where the victim is queried with completely random inputs. While the zero-short learning framework also queries the victim with synthetic images, the queried images are synthesized to maximize the disparity between predictions of the student and teacher. On the contrary, in case of *Random-query*, we query the victim by sampling from a normal distribution, $x \sim \mathcal{N}(0, 1)$. DI is resilient to completely random queries as well.

|       | Threat Model      | ImageNet Architecture | $\Delta\mu$ | p-value     |
|-------|-------------------|-----------------------|-------------|-------------|
| $\mathcal{V}$ | Source    | Wide ResNet-50-2      | 1.868       | $10^{-34}$  |
| $\mathcal{A}_D$ | Diff. Architecture | AlexNet      | 0.790       | $10^{-3}$   |
|       | Diff. Architecture | Inception V3          | 1.085       | $10^{-5}$   |

Table 3: Ownership Tester's effect size in a small-data regime (using only $m = 10$ samples) on the ImageNet dataset using Blind Walk attack.

| Fraction Overlap | $\Delta\mu$ | p-value                  |
|------------------|-------------|--------------------------|
| 0.0              | -0.172      | 0.308                    |
| 0.3              | 0.499       | $7.93 \times 10^{-3}$    |
| 0.5              | 0.514       | $5.78 \times 10^{-3}$    |
| 0.7              | 0.576       | $2.52 \times 10^{-3}$    |
| 1.0              | 0.566       | $3.45 \times 10^{-3}$    |

Table 4: Ownership Tester's effect size in a small-data regime (using only $m = 10$ samples).

Note that we do **not** include *Random-query* as a threat model in case of CIFAR10 and CIFAR100 datasets because random querying is insufficient to achieve model extraction accuracies greater than the majority class baseline in more complicated tasks such as these. However, in case of SVHN, we were able to train an extracted model with 90.2% test set accuracy using random queries alone. Similar observations have been shared in other model extraction literature as well (Truong et al., 2021). We found that our conclusions hold for this additional dataset and we can claim ownership with as few as 10 examples.

**ImageNet.**   We remark that prior work in model extraction has not successfully demonstrated the efficacy of model stealing on large-scale benchmarks like ImageNet. These methods require many queries to steal a model, and are hence not practical yet. As a proof of validity of our approach, we demonstrate dataset inference (DI) in the threat model that assumes complete data theft: the adversary directly steals the dataset used by the victim to train a model rather than querying the victim model. We use 3 pre-trained models on ImageNet using different architectures, and treat the one with a Wide ResNet-50-2 (Zagoruyko & Komodakis, 2016) backbone as the victim. We then observe if DI is able to correctly identify that the two other pre-trained models (AlexNet (Krizhevsky, 2014) and Inception V3 (Szegedy et al., 2015)) were also trained on the same dataset (i.e., ImageNet). We confirm that DI is able to claim ownership (i.e., that the suspect models were indeed trained using knowledge of the victim's training set) with a p-value of $10^{-3}$ on revealing only 10 samples.

For performing DI, the victim trains the confidence regressor with the helps of embeddings generated by querying the Wide ResNet-50-2 architecture over the training and validation sets separately. We first note that the confidence-regressor generalizes well to other points in the victim's train and test set, despite the fact that the ImageNet dataset is orders of magnitude larger than the previous benchmarks experimented on. With only 10 examples, we attain p-values less than $10^{-30}$. Finally, to test if this generalization holds for other architectures that underwent a disjoint training procedure, we experiment over AlexNet and Inception V3. We find that given 10 examples from the ImageNet training dataset, DI can confidently say that the suspect models utilize knowledge of the victim's training set with p-values less than $10^{-4}$. Note that these models are trained on large datasets where works in MI attacks train victims on small subsets of training datasets to enable overfitting Yeom et al. (2018). We believe this demonstrates that DI scales to complex tasks.

## F   EXTENT OF OVERLAP

In this section we elaborate upon the effect of overlap between private datasets of two parties and how does dataset inference respond to such scenarios. More specifically, we study the amount of overlap required for DI to be able to claim theft of common knowledge in the following scenario: We consider a competitor (or adversary) who owns their own private training dataset $\mathcal{S}_\mathcal{A}$. The adversary gains access to the victim's training dataset $\mathcal{S}_\mathcal{V}$. The adversary now trains their ML model

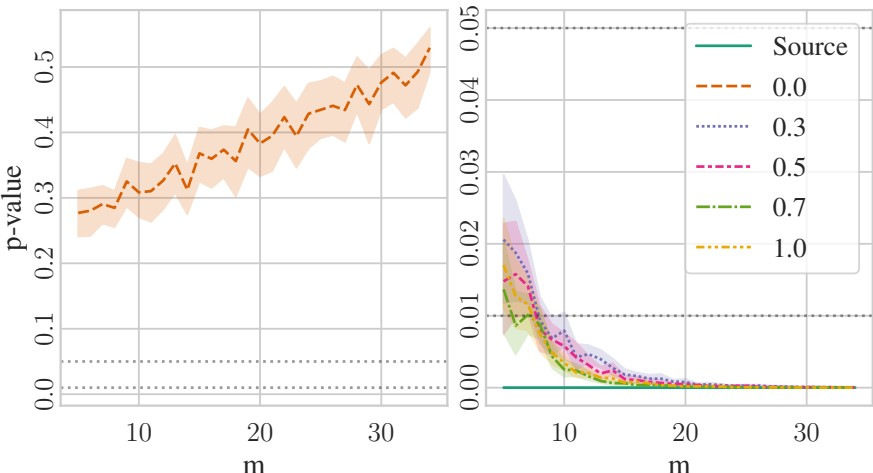

Figure 5: Ownership Tester's p-value depicted as a function of number of training samples revealed ($m$). In the figure to the left, for an honest adversary whose dataset has no overlap with the victim's dataset, the p-value increases as the number of revealed samples increases, indicating decrease in confidence of claim for knowledge theft. For all adversaries with fractional data overlap (to the right), DI is able to achieve a p-value of less than 0.01 in under 10 samples.

on $\mathcal{S} = \mathcal{S}_\mathcal{A} \cup \mathcal{S}_\mathcal{V}^\lambda$, where $\mathcal{S}_\mathcal{V}^\lambda \subset \mathcal{S}_\mathcal{V}$ and $|\mathcal{S}_\mathcal{V}^\lambda| = \lambda |\mathcal{S}_\mathcal{V}|$. That is the new dataset $\mathcal{S}$ has a fraction $\lambda$ of the training points private to $\mathcal{V}$.

Since at the training time the adversary optimizes the 'prediction margin' over all points in $\mathcal{S}$, the prediction margin for points in $\mathcal{S}_\mathcal{V}$ also gets affected. At test time when the victim queries on these points, DI is expected to succeed.

We validate this claim on the SVHN dataset which provides a set of 'extra' images apart from 'train' and 'test' sets. We train the adversary on the union of 'extra' and varying fractions of the 'train' set, where the 'train' set is supposed to be private to the victim. At the time of dataset inference, the victim queries 50 samples from its private 'train' and 'test' set to the adversary's model. Dataset Inference succeeds with p-value = $10^{-3}$ for fractions of overlap = 0.3, 0.5, 0.7, 1.0 as tested. More importantly, as the overlap goes to 0, DI once again does not claim knowledge theft. We present our results for DI on revealing 10 samples in Table 4 and present a graphical illustration of these results with varying numbers of samples revealed in Figure 5. From Table 4, it may also be noted how the effect size increases with the amount of overlap of the private training set, indicating that the DI is becoming increasingly confident of knowledge theft.

