# OpenReview forum: "Dataset Inference: Ownership Resolution in Machine Learning"
_ICLR.cc/2021/Conference — ICLR 2021 Spotlight_

### Official Review · AnonReviewer2 · 2020-10-26
**Interesting work and findings, but current draft needs work**

**Rating:** 7
**Confidence:** 4

**Review:**

**Update** : Since nearly all of my issues have been addressed, I have changed my rating from 5 to 7. Best of luck :)

Summary:

The authors study a subset of model stealing by introducing dataset inference, a process to identify whether a suspected adversary's model has private knowledge from the original model's dataset. Through their experiments on two related tasks in the vision domain (CIFAR-10,100), the authors demonstrate their approach for dataset inference that combines statistical tests and decision boundary-based checks to make claims of model theft. They base their algorithms on theoretical results in toy settings (proofs shifted to additional materials).

##########################################################################

Reasons for score:

Overall, I feel the final observations are quite useful and pave new methods for asserting model theft in real-world settings. Although the proposed techniques are not novel (in fact, a lot of existing literature is not cited, giving the impression that a lot of proposed techniques are contributions of the paper itself), the end results and observations can be useful in practical settings. I feel the authors need to make a lot of changes to the current draft before it can be ready for acceptance.

##########################################################################

Pros:

- Mentioning points such as how issues like these can actually get to courts in real life and are covered by intellectual theft laws is great to see. Since the proposed solution tackles a real-world (potentially) problem, it makes sense to incorporate real-world implications as well.

-  Using theoretical analyses to justify the proposed methods is a great way to approach a problem like this. Testing them out on simpler problems to see how and most importantly, why they work, is important.

- Evaluation over multiple model-stealing attacks is a great way to both show the robustness of the proposed attack method, as well as identify strong (fine-tuned models) as well as weak (label-query attacks) points of the algorithm.

- The query efficiency of the proposed method is quite remarkable. A method that can use these little samples in a real-world setting to assert model stealing (or check for it) with high confidence can be very helpful.

- Page 8: 'gradient-based approaches are sensitive to numerical instabilities' is a great and very valid point. An adversary could in fact even deliberately resort to gradient masking to avoid such an attack.

##########################################################################

Cons:

- By biggest and foremost problem is with the threat model. In all of these settings, it is assumed that the party accusing the adversary of stealing dataset access provides actual data from its own training dataset. However, since this is a real-world scenario, there should also be some way for the accuser to provably show that the data points it is using for the test are in fact from its own training set. Also, it should be made clear what level of overlap in the datasets used is problematic. This parameter would vary from problem to problem and thus cannot be one fixed value. Nonetheless, it should be a parameter of the testing process.

- The term 'dataset inference' is slightly misleading. In some portions, the authors say "we propose to identify stolen copies by showing that they were trained on the same dataset". In other places, they imply that the adversary used a "subset" of that knowledge. It is important to both be consistent in this definition, and to quantify it: nowhere in the paper is it mentioned how much overlap is acceptable as "sheer chance/possible via domain knowledge", and how much is "definitely stealing".

- Key implementation details, such as how the datasets were split to be used by the target and adversary to train their models are omitted.

- CIFAR-10 and CIFAR-100 are highly related datasets (same images). Claims made in the paper would hold more value if they can show similar trends on some other datasets. The authors could try some high-dimensional datasets such as tiny-Imagenet, or even something as simple as MNIST.

- In several places in the paper, authors claim that watermarking models does not work in the sense that trying to copy a model does not copy the watermarks, since there is a difference in distributions. This is not true: recent works have shown honeypot attacks that transfer watermark behavior even when someone tries to copy a model using normal data. Please refer to Section 3.2.2 of [this work](https://arxiv.org/pdf/2009.12153.pdf)

- When using sample points to assert model stealing, does the party arguing for their case release them all together, or one by one. One may argue that the sequential release of sample points could potentially be games (a malicious party could easily look at the target model and accuse them wrongfully). What are the countermeasures against such a scenario? It seems the final judgment can only be made by a third party by the other two parties revealing large subsets of their training sets and algorithms. This might not be possible in real-world scenarios, where datasets used to train models are bound by certain privacy-protection laws (GDPR, etc).

- Section 2 "V suspects theft": what are the criteria here? Is it based on similarity in model performance, or pure suspicion? Since this section sets up the threat model, it might be beneficial to be as exact as possible.
   (a) "may gain access to a subset of". As asked above, how big of a subset is a problem? Since the subset is of Kv and not Sv, it may be possible that they are not overlapping at all, or barely overlap. Would it still be a problem?
   (b) Definition 1: "prove that some knowledge" since this is a definition, it should not be hand-wavy. In my opinion, the "some" here should be quantified perhaps by a parameter that captures the overlap in the two datasets and integrates them in the definition.
   (c) Definition 2: Please rewrite the last point as a conditional assignment. It is hard to read in its current form

- Section 3.3, 'Failure if Membership Inference' refers to two labels: b=1 when the sample is from S, or b=0 when the sample is from D. However, S is a subset of D and thus the first case will always be true. Assuming you mean 'sample is from S - D', please fix this.

- Insufficient literature review. There are many works in the field that deal with the same problem and similar approaches, even though they do not define them as "dataset inference". Please search for some works on 'property inference': the current problem can be posed as differentiating between the dataset suspected to be used, versus some other dataset.

- Although the white-box method looks at finding the 'closest' data points, the black-box method samples random directions and takes steps. Assuming this is intended to be a proxy of the white-box setting, it should take the form of sampling random initial directions and then, over all the samples, picking the closest directions.  Also, the white-box setting may give significantly different results if either of the models were trained with an adversarial robustness objective. It would be interesting to see how the attack performs in that case.

- Section 4.2 claims that membership inference attacks do not take confidence into account and that this is a key difference from the proposed method in this work. This is not true: ROC curves for such attacks can be analyzed to set rejection regions for such claims.

- Section 4.2, last line: "publicly available data that is not used for training of Fv" Is there a specific reason to use data that was not used to train Fv?

- Page 8, 'White-box access is not essential to DI' claims the Blind-Walk approach is 'non-targeted'. This can also be made true for the white-box setting by adjusting the minimization objective to be non-targeted, instead of making it targeted and targeting the top-k classes.

Please address and clarify the cons above

---

> ### Author Response · Authors · 2020-11-19
> **Response 6**
>
> > *** [11] Section 4.2 claims that membership inference attacks do not take confidence into account and that this is a key difference from the proposed method in this work. This is not true: ROC curves for such attacks can be analyzed to set rejection regions for such claims.***
>
> Dataset Inference utilizes statistical confidence measures. Such confidence measures naturally take into account the likelihood of failure because they are evaluated by sampling multiple data points from given distributions. On the contrary, MI works on pre-set rejection thresholds defined using ROC curves. The former can give us the likelihood of failure of dataset inference, whereas the latter is only as representative as the examples tested on for creating the ROC curve. In a new distribution, the generalization of the ROC curve based threshold may or may not hold, but since DI is able to draw multiple samples from the true distribution, we can set statistical confidence regions.
>
> > *** [12] Section 4.2, last line: “publicly available data that is not used for training of Fv” Is there a specific reason to use data that was not used to train Fv?***
>
> The alternate dataset should not have been seen by Fv at training time. Otherwise, the `prediction margin’ for these data points would have been altered during optimization. The confidence regressor would hence not be able to distinguish between the two types of data points (private training and public training), since they were both optimized on. That said, the reviewer is correct that the alternate dataset need not be publicly available. It could for instance be the private test set of the victim.
>
> > *** [13] Page 8, ‘White-box access is not essential to DI’ claims the Blind-Walk approach is ‘non-targeted’. This can also be made true for the white-box setting by adjusting the minimization objective to be non-targeted, instead of making it targeted and targeting the top-k classes.***
>
> The point we wanted to highlight here is that the blind-walk attack is not ranking possible classes based on their distance to the current point (which would be the case even if we used an untargeted minimization objective in the white-box approach) but rather measures the average distance to randomly picked target classes. This follows from our discussion over comment [10].
>
> We have edited the text to:
>
> *“(b) the approach is stochastic and aims to find the expected prediction margin rather than the worst-case (it searches for any incorrect neighboring class in a randomly chosen direction rather than focusing on the distance to possible target classes)”*
>
> ---

---

> > ### Comment · AnonReviewer2 · 2020-11-19
> > **All concerns addressed**
> >
> > Thanks to the authors for such a detailed response. I think most of my concerns have been addressed, to the point where I can revisit the paper with these changes in mind and reevaluate it :)

---

> ### Author Response · Authors · 2020-11-19
> **Response 5**
>
> > ***[8] Section 3.3, 'Failure of Membership Inference' refers to two labels: b=1 when the sample is from S, or b=0 when the sample is from D. However, S is a subset of D and thus the first case will always be true. Assuming you mean 'sample is from S - D', please fix this.***
>
> Thank you for pointing it out. We have fixed this at relevant places in the paper.
>
> ---
>
> > ***[9] Insufficient literature review. There are many works in the field that deal with the same problem and similar approaches, even though they do not define them as "dataset inference". Please search for some works on 'property inference': the current problem can be posed as differentiating between the dataset suspected to be used, versus some other dataset.***
>
> We used the additional page provided at this stage of the revision to move the related work section to the main paper. In addition, we also added a discussion of prior work on property inference in our paper. Thank you for suggesting this relevant comparison.
>
> > ***[10] Although the white-box method looks at finding the ‘closest’ data points, the black-box method samples random directions and takes steps. Assuming this is intended to be a proxy of the white-box setting, it should take the form of sampling random initial directions and then, over all the samples, picking the closest directions. Also, the white-box setting may give significantly different results if either of the models were trained with an adversarial robustness objective. It would be interesting to see how the attack performs in that case.***
>
> The black-box method is not a proxy of the white-box setting in the sense that it uses a different strategy: as you stated the black-box method considers random directions whereas the white-box method looks at the adversarial direction. The empirical benefits of the blind walk attack suggest that attempting an average estimate of the model prediction landscape around a given point, rather than the worst case distance to boundary is more useful for assessing the membership of data points. This can be explained by analyzing what happens during the optimization step of ML models. Whereas on average, each point in the training set makes some contribution to the update of model weights (average case), we know that such models are *not* naturally robust to worst case directions or adversarial examples even for points in the training set. Which means that standard training does not guard models from the adversarial objective, and hence it may not be the most representative estimator of whether a point was optimized over or not.
>
> Following the last statement, if the victim model was trained with an adversarial objective, we would only expect the performance of white-box DI to improve. Indeed, models trained with an adversarial robustness objective were found by [Song et al. (2019)]([https://www.princeton.edu/~pmittal/publications/liwei-dls19.pdf](https://www.princeton.edu/~pmittal/publications/liwei-dls19.pdf)) to be more susceptible to membership inference. The blind walk attack is unaffected by such concerns since it samples random directions.
>
> ---

---

> ### Author Response · Authors · 2020-11-19
> **Response 4**
>
> > ***[6] When using sample points to assert model stealing, does the party arguing for their case release them all together, or one by one. One may argue that the sequential release of sample points could potentially be games (a malicious party could easily look at the target model and accuse them wrongfully). What are the countermeasures against such a scenario? It seems the final judgment can only be made by a third party by the other two parties revealing large subsets of their training sets and algorithms. This might not be possible in real-world scenarios, where datasets used to train models are bound by certain privacy-protection laws (GDPR, etc).***
>
> We expand here on our response to comment [1]. In summary, we need to ensure (a) that the samples released by the victim are their own and (b) that the sequential release of data points would not be problematic; and finally, (c) that the release of the samples does not infringe upon privacy laws.
>
> The first two issues (a) and (b) are addressed if model owners release hashes of their training data along with a model since the sequential release of data points would no longer prove useful in a scenario where training data must be hashed before-hand.
>
> Regarding (c), we note that we can resolve ownership claims with as few as 50 samples in a training set of 50K samples (0.1%) with 99% confidence despite a stringent testing setup. This means few points need to be revealed if the defender wants to assess ownership independently by querying the suspected model themselves. If this is not possible due to privacy-protection laws (GDPR, etc), the defender could involve a third party as you suggested and have the third party perform dataset inference to avoid having to release these few points to the suspected adversary.
>
> In settings where there is no trusted third party available, we believe that it would be interesting to consider how cryptographic primitives could be applied here to provide confidentiality guarantees for the test inputs.
>
> ---
>
> > ***[7] Section 2 "V suspects theft": what are the criteria here? Is it based on similarity in model performance, or pure suspicion? Since this section sets up the threat model, it might be beneficial to be as exact as possible.***
>
> Yes, this is based on suspicion. For instance OpenAI might find it suspicious if an entity releases a language model with similar generalization capabilities to GPT-3 within a few days of OpenAI releasing their GPT-3 model to the public. One could also imagine a tip by an insider of the suspected entity.
>
> > ***(a) "may gain access to a subset of". As asked above, how big of a subset is a problem? Since the subset is of Kv and not Sv, it may be possible that they are not overlapping at all, or barely overlap. Would it still be a problem?***
>
> We covered this part of the comment in our response to comment [1] above.
>
> > ***(b) Definition 1: "prove that some knowledge" since this is a definition, it should not be hand-wavy. In my opinion, the "some" here should be quantified perhaps by a parameter that captures the overlap in the two datasets and integrates them in the definition.***
>
> We agree with the reviewer that quantifying the overlap between the knowledge of the victim and the knowledge contained within a suspect model can be beneficial. However, we want to point out that an overlap between training sets is only a particular case of an overlap in knowledge. For instance, in the case of model extraction, it is likely that the stolen model’s training set has 0 overlap with the victim's training set. Therefore, we argue that the meaning of knowledge overlap is largely dependent on the threat model, and is often not trivial to measure. Furthemore, we argue that the natural number “m” (the number of revealed samples) and the False Positive Error rate “alpha” can provide a bound for the knowledge overlap which can be tolerated. In other words, for a given m and alpha, the maximum overlap between victim’s and adversary’s knowledge is bounded. We have carried out a new set of experiments (See Appendix E in the updated submission) to empirically quantify this relationship. Our results show that increasing the possible knowledge overlap (in this case, training set overlap) requires us to increase m, or otherwise set a higher alpha.
>
> > ***(c) Definition 2: Please rewrite the last point as a conditional assignment. It is hard to read in its current form***
>
> We have updated the paper accordingly.
>
> ---

---

> ### Author Response · Authors · 2020-11-19
> **Response 3**
>
> > ***[4] CIFAR-10 and CIFAR-100 are highly related datasets .... try some high-dimensional datasets such as tiny-Imagenet, or even something as simple as MNIST.***
>
> Thank you for the suggestion. We added experiments on SVHN and ImageNet datasets to provide supplemental evidence for the general applicability of dataset inference. The results are described in Appendix E, and also repeated below:
>
> For SVHN, we also introduce a new threat model **Random-Query** which uses totally random inputs sampled from N(0,1) to perform model extraction. The results for DI using the Blind Walk attack follow similar trends to those in the main paper, as can be seen below:
>
>          Dataset Inference on SVHN (Blind Walk)
>
> | Attack Type     | Model Stealing Attack  | Mean Difference | p-value |
> |-----------------|------------------------|-----------------|---------|
> | $\mathcal{V}$   | Source                 |     1.543            |    $10^{-19}$     |
> | $\mathcal{A_D}$ | Distillation           |       0.687           |   $10^{-04}$      |
> | $\mathcal{A_D}$ | Different Architecture |      0.495           |    $10^{-03}$      |
> | $\mathcal{A_M}$ | Zero-Shot Learning     |      0.637           |   $10^{-03}$      |
> | $\mathcal{A_M}$ | Fine-tuning    |       0.688            |   $10^{-04}$      |
> | $\mathcal{A_Q}$ | Label-query            |     0.566             |  $10^{-03}$       |
> | $\mathcal{A_Q}$ | Logit-query            |     0.639            |   $10^{-03}$      |
> | $\mathcal{A_Q}$ | **Random-Query**                 |     0.539            |   $10^{-03}$      |
> | $\mathcal{I}$   | Independent            |         -0.363 |  $0.567$           |
>
>
> For ImageNet, we first remark that prior work in model extraction has not successfully demonstrated the efficacy of model stealing on such complex tasks, since these methods require a high number of queries to steal a model, and are hence not practical yet. As proof of validity of our approach, we demonstrate DI in the threat model that assumes complete data theft: the adversary directly steals the dataset used by the victim to train a model. We use 3 pre-trained models on ImageNet using different architectures, and treat the one with a Wide ResNet-50-2 backbone as the victim. We then observe if DI is able to correctly identify that the other pre-trained models (AlexNet & Inception V3) were also trained on the same dataset (ImageNet). We confirm that DI is able to claim ownership (i.e., that the suspect models were trained using knowledge of the victim’s training set) with a p-value of $10^{-4}$ in less than 10 samples. The results are provided below:
>
>          DI on ImageNet (Blind Walk)
>
> | Attack Type     | Model Stealing Attack  | Architecture | Mean Difference | p-value |
> |-----------------|------------------------|-----------|------|---------|
> | $\mathcal{V}$   | Source                   |    Wide Resnet-50-2    |    1.868  | $10^{-34}$        |
> | $\mathcal{A_D}$ |  Diff. Architecture    |    AlexNet       |        0.790 | $10^{-3}$       |
> | $\mathcal{A_D}$ | Diff. Architecture      |   Inception V3       |    1.085 |	$10^{-5}$        |
>
> Recall that these models are trained on large datasets where works in MI attacks train victims on small subsets of training datasets to enable overfitting (Yeom et. al. 2018). We believe this demonstrates that DI scales to complex tasks. These results were added in Appendix E of the updated draft.
>
> ---
>
> > ***[5] ...authors claim that watermarking models does not work in the sense that trying to copy a model does not copy the watermarks...recent works have shown honeypot attacks that transfer watermark behavior ...Please refer to Section 3.2.2 of [this work](https://arxiv.org/pdf/2009.12153.pdf)***
>
> A key difference between our approach and watermarking techniques is that watermarking involves modifications to the training procedure whereas we can apply dataset inference to any model already trained and deployed. Further, watermarking has a trade-off on task accuracy.
>
> Regarding the effectiveness of watermarking, we cited Jia et al (2020) where it is shown that common watermarking techniques fail in the presence of model extraction via distillation. This is because the distribution of watermarks differs from the distribution of the data that is being modeled. This is also supported by Yang et al. (2019) for the particular case of distillation.
>
> We agree with the reviewer that a more substantiated and scoped claim is desirable here. Therefore, we qualified our claim about the limited effectiveness of the watermarking technique with the caveat that both Jia et al. (2020) and Yang et al. (2019), have proposed alternative watermarking strategies that prevail distillation and fine-tuning (to some extent—see the next response). We made this qualification more prominent in the updated submission.
>
> We kindly ask the reviewer to clarify the term “honeypot attack” in the context we consider. We could not find the term honeypot in the survey cited.
>
> ---

---

> ### Author Response · Authors · 2020-11-19
> **Response 2**
>
> > ***[2] The term 'dataset inference' is slightly misleading. In some portions, the authors say "we propose to identify stolen copies by showing that they were trained on the same dataset". In other places, they imply that the adversary used a "subset" of that knowledge. It is important to both be consistent in this definition, and to quantify it: nowhere in the paper is it mentioned how much overlap is acceptable as "sheer chance/possible via domain knowledge", and how much is "definitely stealing".***
>
> **Definitions**
>
> We chose the term dataset inference to reflect the fact that our defense mechanism relies on the fact that the defender is able to infer whether or not a model used knowledge originally contained in the defender’s dataset. This encompasses multiple scenarios:
>
> - In the broadest definition of model stealing possible, an adversary may use information (knowledge) contained within the victim's trained model to create a stolen model. A concrete example is distillation.
> - It is also possible that a subset of the victim's private training set is leaked online. In that case, the adversary uses actual data samples from the victim's training set to create the stolen model.
>
> We showed that our framework can successfully detect attackers in both settings, which is why we stated in our manuscript in Introduction (paragraph 3) that we can detect partial (e.g. an actual subset of the dataset) and indirect (e.g. distillation) use of the victim’s dataset: “we propose to identify stolen copies by showing that they were trained (at least partially and indirectly) on the same dataset as the victim model.”
>
> In the updated submission, we have strived to make this clear and consistent throughout the paper.
>
> **On Overlap**: Please refer to response to comment [1]
>
> ---
>
> > ***[3] Key implementation details, such as how the datasets were split to be used by the target and adversary to train their models are omitted.***
>
> We used the full training data to train the victim model. The exact implementation details for training the adversary's model were deferred to Appendix C as remarked in Section 5.2. **We have moved the complete information to the main paper in Section 5.2**, and included the corresponding extract below:
>
> *“””*
>
> *For model extraction and fine-tuning attacks, we use a subset of 500,000 unlabeled TinyImages that are closest to CIFAR10, as created by Carmon et. al. (2019). More details about the creation of the dataset can be found in their work.*
>
> *For CIFAR100, we use the STL-10 dataset to steal the models. We train the student model for 20 epochs in the case of model extraction methods and 5 epochs for fine-tuning. For Zero-shot learning, we use the data-free adversarial distillation method proposed by Fang et. al. (2019) and train the student model for 200 epochs. In the case of distillation and modified architecture, we have access to the original training data of the victim. We train both models for 100 epochs on the full training set.*
>
> *In all the training methods, we use a fixed learning rate strategy with SGD optimizer and decay the learning rate by a factor of 0.2 at the end of the $0.3\times$, $0.6\times$, and $0.8\times$ the total number of epochs.*
>
> *“””*
>
> As an aside, we also remark that using the complete training sets ensures that the victim does not overfit on the training data. This is contrary to membership inference works which train on small subsets of the training data, in order to overfit and be able to stage successful MI attacks. Our work is hence, more generally applicable to real-world scenarios.
>
> ---

---

> ### Author Response · Authors · 2020-11-19
> **Response 1**
>
> We would like to thank you for the thorough feedback, we were working on editing the paper to reflect comments made in your review. The changes made in our revised manuscript are detailed in our responses below. We numbered your list of “Cons” to simplify cross-referencing our answers. We are happy to answer any remaining questions you may have.
>
> ---
>
> > ***[1] My biggest and foremost problem is with the threat model. In all of these settings, it is assumed that the party accusing the adversary of stealing dataset access provides actual data from its own training dataset. However, since this is a real-world scenario, there should also be some way for the accuser to provably show that the data points it is using for the test are in fact from its own training set. Also, it should be made clear what level of overlap in the datasets used is problematic. This parameter would vary from problem to problem and thus cannot be one fixed value. Nonetheless, it should be a parameter of the testing process.***
>
> - **Practical Considerations**
>
> This is a great point: in summary, we need to ensure that the samples released by the victim are their own. This falls under the realm of a more traditional computer security problem for which we propose to use existing non-ML mechanisms. To address this, we for instance propose to require that all model owners --- who wish to maintain their right to claim ownership later through dataset inference --- publish hashed values of their training data samples. When a claim is made, the legal authority can easily check whether the hash of the released samples match one of the previously published hashes, or not. This will ensure that the dataset can not be altered to falsely malign an honest competitor.
>
> - **Amount of Overlap**
>
> We study the amount of overlap required for DI to be able to claim theft of common knowledge in the following scenario: We consider a competitor (or adversary) who owns their own private training dataset $\mathcal{S_A}$. The adversary gains access to the victim’s training dataset $\mathcal{S_V}$. The adversary now trains their ML model on $\mathcal{S} = \mathcal{S_A}$ U  $\mathcal{S_V}^{\lambda}$, where $\mathcal{S_V}^{\lambda} \subset \mathcal{S_V}$ and $|\mathcal{S_V}^{\lambda}| = \lambda |\mathcal{S_V}|$. That is the new dataset $\mathcal{S}$ has a fraction $\lambda$ of the training points private to $\mathcal{V}$.
>
> Since at the training time the adversary optimizes the 'prediction margin’ over all points in $\mathcal{S}$, the prediction margin for points in $\mathcal{S_V}$ also gets affected. At test time when the victim queries on these points, DI is expected to succeed.
>
> We validate this claim on the SVHN dataset which provides a set of ‘extra’ images apart from ‘train’ and ‘test’ sets. We train the adversary on the union of ‘extra’ and varying fractions of the ‘train’ set, where the ‘train’ set is supposed to be private to the victim. At the time of dataset inference, the victim queries 50 samples from its private ‘train’ and ‘test’ set to the adversary’s model. Dataset Inference succeeds with p-Value = 1e-3 for fractions of overlap = 0.3, 0.5, 0.7, 1.0 as tested. More importantly, as the overlap goes to 0, DI once again does not claim knowledge theft. Please refer to a graphical representation of these results with varying numbers of samples revealed in Appendix F. We present the table of results for convenience below:
>
>     Performance of DI with varying degree of data overlap
>
> | Fraction Overlap | $\Delta \mu$ | p-Value |
> |-----------------|------------------------|--------------------------|
> | 0.0 | -0.172 | 	$0.308$  	|
> | 0.3 | 0.499	| 	$7.93\times10^{-3}$  	|
> | 0.5 | 0.514 | 	$5.78\times10^{-3}$  	|
> | 0.7 | 0.576 | 	$2.52\times10^{-3}$  	|
> | 1.0 | 0.566 | 	$3.45\times10^{-3}$  	|
>
> From the table, it may also be noted how the effect size increases with the amount of overlap of the private training set, indicating that the DI is becoming increasingly confident of knowledge theft. We further discuss the nature of overlap and whether it can be quantized in our response to comment [7.b].
>
> ---

---

### Official Review · AnonReviewer1 · 2020-10-28
**the proposed dataset inference approach is novel**

**Rating:** 7
**Confidence:** 4

**Review:**

This paper tackles a timely problem of detecting model stealing attacks. The proposed identifies the stolen model by investigating whether the model is trained on the same dataset as the victim model.

Pros:
-	The proposed method of dataset inference for model stealing detection is novel.
-	The experiments provided by the paper are comprehensive, including different assumptions of data access, model access, and query access, etc. Adaptive attacks are considered in the paper as well.
-	The paper is well-written and easy to follow.

Cons:
-	Zero-shot learning ([9] in the paper) cannot be directly used as a model stealing attack, because the proposed approach in [9] requires more information of the victim model than the input and output pair. It would be better if the paper could investigate other data-free model stealing approaches (e.g., [1][2]). Also, it is not clear to me how the proposed dataset inference approach is applied to data-free model stealing attacks, since the training dataset used by the attacker is different from the victim’s dataset. If the attackers use some synthetic data to steal the model, will the proposed approach work?
-	What are the white-box and black-box settings in Section 4.1?


[1] Mika Juuti, Sebastian Szyller, Samuel Marchal, and N Asokan. Prada: protecting against dnn model stealing attacks. In 2019 IEEE European Symposium on Security and Privacy (EuroS&P), pp. 512–527. IEEE, 2019.

[2] Nicolas Papernot, Patrick McDaniel, Ian Goodfellow, Somesh Jha, Z Berkay Celik, and Ananthram Swami. Practical black-box attacks against machine learning. In Proceedings of the 2017 ACM on Asia conference on computer and communications security, pp. 506–519, 2017.

---

> ### Author Response · Authors · 2020-11-19
> **Clarifications on Threat Model (Response 2)**
>
> > ***Zero-shot learning ([9] in the paper) cannot be directly used as a model stealing attack, because the proposed approach in [9] requires more information of the victim model than the input and output pair***
>
> As detailed in the introduction (paragraph 2, point **(3)**) and Section 5.2 (point (2)) on the type of threat models, our work on dataset inference extends beyond the realms of model extraction alone. We also consider the following two scenarios:
>
> - complete data theft (i.e., the training dataset itself is leaked to the adversary)
> - model availability: in the age of open-sourced code and pre-trained models, the threat of commercialization of an ML model, otherwise intended only for academic purposes, is practical and prevalent. For instance, consider the scenarios where a company (such as OpenAI) may want to open-source their model (like GPT-3) for academic purposes but does not permit its commercialization.
>
> In both situations, an adversary has sufficient information for staging zero-shot learning attacks on the victim model. An additional advantage of considering zero-shot learning is that it allows us to strengthen our evaluation with an analysis of dataset inference against adaptive adversaries: the queries used in zero-shot learning negate any resemblance to real-world queries since they only utilize synthetic inputs.
>
> ---
>
> > ***What are the white-box and black-box settings in Section 4.1?***
>
> The white-box and black-box settings indicate access levels that the victim has to the potential adversary’s model. For instance, if the victim suspects that a competitor’s API has stolen their ML model, they can only gain black-box access to the predictions of their MLaaS. However, if the victim files a legal complaint, then in the presence of a neutral arbitrator, the victim may be presented white-box access to demonstrate ‘dataset inference’ on the suspect adversary. Section 4.1 introduces an embedding generation procedure for each of the two settings.
>
> ---
>
> Thank you for your time. We would be happy to answer any further queries.

---

> ### Author Response · Authors · 2020-11-19
> **On synthetic query generation for model extraction (Response 1)**
>
> We thank you for your insightful comments and careful analysis of our paper. We will address your concerns in a sequence of two comments.
>
> ---
>
> > ***It would be better if the paper could investigate other data-free model stealing approaches (e.g., [1][2])***
>
> > ***If the attackers use some synthetic data to steal the model, will the proposed approach work?***
>
> Thank you for the suggestion. We would like to clarify some details about the goals and problem settings of [1],[2] (in the review):
>
> 1. The authors of [1] propose two techniques for synthetic sample generation but these samples are still derived from a dataset of real and labeled data. For example in Section IV-D for (1) “Jacobian-based Synthetic Sample Generation,” new adversarial examples are crafted starting from original data points; and in (2) “Random Synthetic Sample Generation,” the color channel of each pixel of the images in the original dataset are randomly perturbed. Hence, these techniques are not truly data-free since they require that the adversary bootstrap the attack with a dataset of non-synthetic samples.
> 2. As noted by the authors in [2], the key goal of their work is to achieve enhanced transferability of adversarial examples in the black-box setting, and the accuracy of the substitute classifier is not of importance. See section 4.1 of [2]: “*We emphasize that this technique is not designed to maximize the substitute DNN’s accuracy but rather ensure that it approximates the oracle’s decision boundaries with few label queries*”.
>
> This is why we did not evaluate these attacks in our work and instead chose to evaluate the zero-shot learning attack of Fang et. al. (2019). This attack uses completely synthetic data to steal the victim’s model (see Figure 3 in [Fang et. al. (2019)]([https://arxiv.org/pdf/1912.11006.pdf](https://arxiv.org/pdf/1912.11006.pdf)) for a representative set). However, the predictions of the victim on these synthetic inputs are still a byproduct of the original training data. Thus, dataset inference is still able to capture the desired characteristics of ‘prediction margin’ for training vs. unseen data points and hence claim ownership. We have made this contribution of our work more explicit in our discussion of threat models in Section 5.2.
>
> That said, we agree with your comment that more evaluation of the data-free model stealing setting is valuable. Thus, our revised manuscript adds an evaluation of dataset inference against completely synthetic/random queries. In our new experiments, we found that model extraction based on random querying is able to achieve nontrivial accuracies only on tasks like MNIST and SVHN (e.g., it failed for CIFAR10). This observation is also supported by prior work ([MAZE]([https://arxiv.org/abs/2005.03161](https://arxiv.org/abs/2005.03161))). Therefore, we extracted an SVHN model using random queries, and used dataset inference to determine if this model had been stolen. We found that our conclusions hold for this new threat model and we can claim ownership with as few as 10 examples. The p-values are listed below (all significant for our significance level 0.01). These results were added in our revised manuscript in Appendix E.
>
>                                          Dataset Inference on SVHN (Blind Walk)
>
> | Attack Type     | Model Stealing Attack  | Mean Difference | p-value |
> |-----------------|------------------------|-----------------|---------|
> | $\mathcal{V}$   | Source                 |     1.543            |    $10^{-19}$     |
> | $\mathcal{A_D}$ | Distillation           |       0.687           |   $10^{-04}$      |
> | $\mathcal{A_D}$ | Different Architecture |      0.495           |    $10^{-03}$      |
> | $\mathcal{A_M}$ | Zero-Shot Learning     |      0.637           |   $10^{-03}$      |
> | $\mathcal{A_M}$ | Fine-tuning    |       0.688            |   $10^{-04}$      |
> | $\mathcal{A_Q}$ | Label-query            |     0.566             |  $10^{-03}$       |
> | $\mathcal{A_Q}$ | Logit-query            |     0.639            |   $10^{-03}$      |
> | $\mathcal{A_Q}$ | **Random-Query**                 |     0.539            |   $10^{-03}$      |
> | $\mathcal{I}$   | Independent            |         -0.363 |  $0.567$           |
>
> ---
>
> *Continued in next comment

---

### Official Review · AnonReviewer3 · 2020-10-29
**interesting**

**Rating:** 7
**Confidence:** 4

**Review:**

The submission proposes to defend against model stealing attacks by dataset inference. It is argued that other watermarking techniques are not robust enough to be transferred during model stealing.

Ownership resolution is performed by kind of a membership inference attack that the model owner runs on the potential copy. As only the owner knows the true training data, the "attack" can be amplified by a few samples. This leads to a rather strong signal in order to show model provenance. Final decision is formulated as a hypothesis test in order to get a confidence. The argumentation why for larger models this is not dependent on some sort of over fitting is not clear.

While the idea is interesting, it is somewhat at odds with others goals in machine learning. In general, it is not a desirable property if the trained model contains "artifacts" related to the training data. This on the one hand could be overfitting artifacts that hamper generalization or more prominently could mean that the model does not preserve privacy. If such overfitting is not a major issues and privacy is not a concern, the approach seems viable. However, there should be an extended discussion under which situations/models the method is feasible. (e.g. privacy preserving learning?)

I somewhat disagree with the practice of moving the related work to the appendix. this is a vital part of a paper and not an appendix. this looks like breaking the page limitation. overall, having a 19 page submission is not quite following the submission guideline (although the additional material is useful). this should not become practice - in particular as some parts in the main submission could be more concise. In particular, as the actual algorithm is quite simple (which I don't want to see as a disadvantage as it's well motivated)

Overall, the paper is insightful -- also adding to the understanding of the connections to membership inference. Also the attention to important details like query efficiency is taken care of.

It is bit disappointing that the method is not evaluated on more complex classification problems. In particular, as membership inference (which this method is based on) can be even weaker on those larger and well trained models. The claim about other watermarking techniques no being effective - although probably true - was not substantiated (unless I've missed). at least there is no comparison to other attribution/provenance techniques.

While this submission is a very interesting and enjoyable read, in the end it fails to fully convince how large the impact will be. Questions are open w.r.t. models with privacy guarantees and larger models  ... and also in the end, this technique is still effected by fine tuning. It is unclear to what extend this is an issue.

---

> ### Author Response · Authors · 2020-11-19
> **Response 2**
>
> > ***Not evaluated on more complex classification problems. In particular, membership inference (which this method is based on) can be even weaker on those larger and well-trained models.***
>
> We followed your suggestion and supplemented our paper with results showing the success of *dataset inference* on large models trained on complex tasks like the ImageNet dataset. We first remark that prior work in model extraction has not successfully demonstrated the efficacy of model stealing on such complex tasks, since these methods require a high number of queries to steal a model, and are hence not practical yet. As a proof of validity of our approach, we demonstrate dataset inference (DI) in the threat model that assumes complete data theft: the adversary directly steals the dataset used by the victim to train a model rather than querying the victim model. We use 3 pre-trained models on ImageNet using different architectures, and treat the one with a Wide ResNet-50-2 backbone as the victim. We then observe if DI is able to correctly identify that the two other pre-trained models (AlexNet and Inception V3) were also trained on the same dataset (i.e., ImageNet). We confirm that DI is able to claim ownership (i.e., that the suspect models were indeed trained using knowledge of the victim’s training set) with a p-value of $10^{-3}$ in less than 10 samples. The results are provided below:
>
>      Success of DI on ImageNet (Blind Walk)
>
> | Attack Type     | Model Stealing Attack  | Architecture | Mean Difference | p-value |
> |-----------------|------------------------|-----------|------|---------|
> | $\mathcal{V}$   | Source                   |    Wide Resnet-50-2    |    1.868  | $10^{-34}$        |
> | $\mathcal{A_D}$ |  Different Architecture    |    AlexNet       |        0.790 | $10^{-3}$       |
> | $\mathcal{A_D}$ | Different Architecture      |   Inception V3       |    1.085 |	$10^{-5}$        |
>
> Recall that these models are trained on large datasets where works in MI attacks train victims on small subsets of training datasets to enable overfitting (Yeom et. al. 2018). We believe this demonstrates that DI scales to complex tasks. These results were added in Appendix E of the updated draft.
>
> ---
>
> > ***The claim about other watermarking techniques not being effective - although probably true - was not substantiated (unless I’ve missed). at least there is no comparison to other attribution/provenance techniques.***
>
> A key difference between our approach and watermarking techniques is that watermarking involves modifications to the training procedure whereas we can apply dataset inference to any model already trained and deployed. Further, watermarking has a trade-off on task accuracy.
>
> Regarding the effectiveness of watermarking, we cited [Jia et al (2020)](http://arxiv.org/abs/2002.12200) where it is shown that common watermarking techniques fail in the presence of model extraction via distillation. This is because the distribution of watermarks differs from the distribution of the data that is being modeled. This is also supported by  [Yang et al. (2019)](http://arxiv.org/abs/1906.06046) for the particular case of distillation.
>
> We agree with the reviewer that a more substantiated and scoped claim is desirable here. Therefore, we qualified our claim about the limited effectiveness of the watermarking technique with the caveat that both [Jia et al. (2020)](http://arxiv.org/abs/2002.12200) and [Yang et al. (2019)](http://arxiv.org/abs/1906.06046), have proposed alternative watermarking strategies that prevail distillation and fine-tuning (to some extent—see the next response). We made this qualification more prominent in the updated submission.
>
> ---
>
> > ***...this technique is still effected by fine tuning.***
>
> We would like to clarify that fine-tuning is the least effective attack against DI among the different threat models we considered in our paper. The effect size for fine-tuning is the largest across our experiments (as highlighted in red in Table 1). This is also indicated by the p-values which are correspondingly the lowest for fine-tuning. We have made this more explicit in the table caption. These two observations mean that our DI method was most resilient against fine-tuning based attacks.
>
> ---
>
> Thank you for your time. We would be happy to answer any further queries.

---

> ### Author Response · Authors · 2020-11-19
> **Response 1**
>
> We thank the reviewer for the insightful feedback.
>
> > ***The argumentation why for larger models this is not dependent on some sort of overfitting is not clear***
>
> We would like to clarify that the claim made in the contribution section, and subsequently validated through our analysis, is that dataset inference (DI) is independent of dataset size rather than of model size. All our proofs are made over the same fixed architecture. However, if the dataset size increases (given a fixed model size), the success of membership inference (MI) decreases as proved in Theorem 2 (because overfitting decreases). The same does not happen for DI as the number of samples we can apply the statistical tests to increases with the size of the dataset.
>
> In line with your intuition, we also suspect that the success of DI should increase given a larger model for a fixed data size. In fact, in Section 4.1 we even mention that:
>
> “””
>
> *While we only discuss the case of a linear network in this analysis, the success of DI only increases with the number of parameters in a machine learning model, as is the case for MI (Yeom et al., 2018), which in effect makes the following analysis a stronger result to prove. Prior works have also argued how over-parametrized deep learning networks memorize training points (Zhang et al., 2016; Feldman, 2019). At its core, DI builds on the premise of input memorization, albeit weak.*
>
> “””
>
> ---
>
> > ***This, on the one hand, could be overfitting artifacts that hamper generalization or more prominently could mean that the model does not preserve privacy. If such overfitting is not a major issue and privacy is not a concern, the approach seems viable. However, there should be an extended discussion under which situations/models the method is feasible. (e.g. privacy-preserving learning?)***
>
> It is true that differential privacy (DP) reduces information leakage. However, as shown by [Leino et al (2019)](https://arxiv.org/abs/1906.11798), the epsilons required to render membership inference (MI) ineffective would significantly decrease the model's performance, and larger epsilon (e.g, 16) values sometimes used in the industry, fail to provide any additional defense against MI.
>
>
> We argue that since dataset inference (DI) amplifies the membership signal using multiple private samples, it follows that the epsilon values required to make DI ineffective would be even lower than it is for MI. Therefore, epsilons that can make the model private likely do not interfere with dataset inference. Studying further trade-offs between dataset inference and privacy-preserving learning is a promising future direction.
>
> ---
>
> > ***I somewhat disagree with the practice of moving the related work to the appendix. this is a vital part of a paper and not an appendix.***
>
> We agree with the reviewer, and have moved the our related work section back to the main paper in our revised submission (See Section 2). We also note that we strived to clearly position our work with respect to the most relevant literature, in particular membership inference, in the introduction. The related work section in the appendix aims to expand on that.
>
> ---
>
> *Continued

---

### Decision · Program_Chairs · 2021-01-07
**Final Decision**

**Decision:**

Accept (Spotlight)

**Comment:**

This paper proposed to defend against model stealing attacks by dataset inference. The paper received unanimous rating of "Good paper" and "accept". The reviewers praise this paper insightful and well written. There are active discussion between the reviewers and authors, which further clarify some of the issues. Given the positive review and overall rating, the AC recommends it to be an spotlight paper.